# DEPENDENCY STRUCTURE DISCOVERY FROM INTERVENTIONS

## ABSTRACT

Promising results have driven a recent surge of interest in continuous optimization methods for Bayesian network structure learning from observational data. However, there are theoretical limitations on the identifiability of underlying structures obtained from observational data alone. Interventional data provides much richer information about the underlying data-generating process. However, the extension and application of methods designed for observational data to include interventions is not straightforward and remains an open problem. In this paper we provide a general framework based on continuous optimization and neural networks to create models for the combination of observational and interventional data. The proposed method is applicable even in the challenging and realistic case that the identity of the intervened upon variable is unknown. We examine the proposed method in the setting of graph recovery both de novo and from a partially-known edge set. We establish strong benchmark results on several structure learning tasks, including structure recovery of both synthetic graphs as well as standard graphs from the Bayesian Network Repository.

## 1 INTRODUCTION

Structure learning concerns itself with the recovery of the graph structure of Bayesian networks (BNs) from data samples. A natural application of Bayesian networks is to describe cause-effect relationships between variables. In that context, one may speak of *causal structure learning*. Causal structure learning is challenging because purely observational data may be satisfactorily explained by multiple Bayesian networks (a *Markov equivalence class*), but only one is the most robust to distributional shifts: The one with the correct graph. A more powerful tool than BNs is thus needed to model causal relationships.

Structural Causal Models (SCMs) are that tool. An SCM over a set of random variables is a collection of assignments to these variables and a directed acyclic graph of dependencies between them (Peters et al., 2017, §6.2). Each assignment is a function of only the *direct causes* of a variable, plus an independent noise source. An SCM entails precisely one (observational) data distribution. *Interventions* on an SCM's assignments, such as setting a random variable to a fixed value (*a hard intervention*), entail new *interventional* data distributions (Peters et al., 2017, §6.3).

SCMs can be used to answer higher-order questions of cause-and-effect, up the ladder of causation (Pearl & Mackenzie, 2018). Causal structure learning using SCMs has been attempted in several disciplines including biology (Sachs et al., 2005; Hill et al., 2016), weather forecasting (Abramson et al., 1996) and medicine (Lauritzen & Spiegelhalter, 1988; Korb & Nicholson, 2010).

Causal structure is most frequently learned from data drawn from observational distributions. Structure learning methods generally cannot do more than identify the causal graph up to a Markov equivalence class (Spirtes et al., 2000). In order to fully identify the true causal graph, a method must either make restrictive assumptions about the underlying data-generating process, such as linear but non-Gaussian data (Shimizu et al., 2006), or must access enough data from outside the observational distribution (i.e., from interventions).

Under certain assumptions about the number, diversity, and nature of the interventions, the true underlying causal graph is always identifiable, given that the method knows the intervention performed (Heckerman et al., 1995). In much of the prior work on causal model induction it is assumed that

there is an experimenter and this experimenter performs interventions. However, in the real world, interventions can also be performed by other agents, which could lead to *unknown interventions* (interventions with unknown target variables). A few works have attempted to learn structures from unknown-intervention data (Eaton & Murphy, 2007a; Squires et al., 2020; Huang et al., 2020). A notable such work, (Mooij et al., 2016), has been extended in (Kocaoglu et al., 2019; Jaber et al., 2020). Although there is no theoretical guarantee that the true causal graph can be identified in that setting, evidence so far points to that still being the case.

Another common setting is when the graph structure is partially provided, but must be completed. An example is protein structure learning in biology, where we may have definitive knowledge of some causal edges in the protein-protein interactome, but the remaining causal edges must be discovered. We will call this setting "partial graph completion". This is an easier task compared to learning the entire graph, since it limits the number of edges that have to be learned.

Recently, a flurry of work on structure learning using continuous optimization methods has appeared (Zheng et al., 2018; Yu et al., 2019). These methods operate on observational data and are competitive with other methods. Because of the theoretical limitations on identification from purely observational data cited above, it would be interesting to extend these methods to interventional data. However, it is not straightforward to apply continuous optimization methods to structure learning from interventional data. Our key **contributions** are to answer the following questions experimentally:

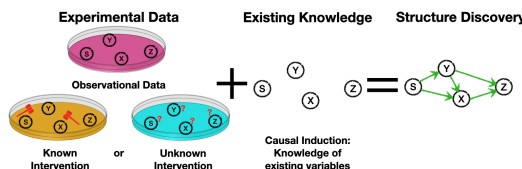

Figure 1: In many areas of science, such as biology, we try to infer the underlying mechanisms and structure through experiments. We can obtain observational data plus interventional data through known (e.g. by targeting a certain variable) or unknown interventions (e.g. when it is unclear where the effect of the intervention will be). Knowledge of existing edges e.g. through previous experiments can likewise be included and be considered a special case of causal induction.

1. Can the proposed model recover true causal structure? Yes, see Figure §4.
2. How does the proposed model compare against state of the art causal methods on real-world datasets? *Favourably;* see §5.4 and Table §1.
3. Does a proposed model generalize well to unseen interventions? Yes, see §5.5.
4. How does the proposed model perform on partial graph recovery? It scales to $\sim 50$ variables while the other baselines can't. see §5.7.

## 2 PRELIMINARIES

**Causal modeling.** A Structural Causal Model (SCM) (Peters et al., 2017) over a finite number $M$ of random variables $X_i$ is a set of structural assignments

$$X_i := f_i(X_{pa(i,C)}, N_i), \quad \forall i \in \{0, \dots, M-1\} \tag{1}$$

**Identifiability.** In a purely-observational setting, it is known that causal graphs can be distinguished only up to a Markov equivalence class. In order to identify the true causal graph structure, interventional data is needed (Eberhardt et al., 2012).

**Interventions.** There are several types of common *interventions* which may be available (Eaton & Murphy, 2007b). These are: *No intervention:* only observational data is obtained from the ground truth model. *Hard/perfect:* the value of a single or several variables is fixed and then ancestral sampling is performed on the other variables. *Soft/imperfect:* the conditional distribution of the variable on which the intervention is performed is changed. *Uncertain:* the learner is not sure of which variable exactly the intervention affected directly. Here we make use of soft intervention because they include hard intervention as a limiting case and hence are more general.

**Structure discovery using continuous optimization.** Structure discovery is a super-exponential search problem that searches though all possible directed acyclic graphs (DAGs). Previous continuous-optimization structure learning works (Zheng et al., 2018; Yu et al., 2019; Lachapelle et al., 2019) mitigate the problem of searching in the super-exponential set of graph structures by considering the degree to which a hypothesis graph violates "DAG-ness" as an additional penalty to be optimized. If there are $M$ such variables, the strategy of considering all the possible structural graphs as separate hypotheses is not feasible because it would require maintaining $O(2^{M^2})$ models of the data.

## 3 RELATED WORK

The recovery of the underlying structural causal graph from observational and interventional data is a fundamental problem (Pearl, 1995; 2009; Spirtes et al., 2000). Different approaches have been studied: score-based, constraint-based, asymmetry-based and continuous optimization methods. Score-based methods search through the space of all possible directed acyclic graphs (DAGs) representing the causal structure based on some form of scoring function for network structures (Heckerman et al., 1995; Chickering, 2002; Tsamardinos et al., 2006; Hauser & Bühlmann, 2012; Goudet et al., 2017; Cooper & Yoo, 1999; Zhu & Chen, 2019). Constraint-based methods (Spirtes et al., 2000; Sun et al., 2007; Zhang et al., 2012; Monti et al., 2019; Zhu & Chen, 2019) infer the DAG by analyzing conditional independences in the data. Eaton & Murphy (2007c) use dynamic programming techniques to accelerate Markov Chain Monte Carlo (MCMC) sampling in a Bayesian approach to structure learning for discrete variable DAGs. Peters et al. (2016); Ghassami et al. (2017); Rojas-Carulla et al. (2018) exploit invariance across environments to infer causal structure, which faces difficulty scaling due to the iteration over the super-exponential set of possible graphs. Recently, (Zheng et al., 2018; Yu et al., 2019; Lachapelle et al., 2019) framed the structure search as a continuous optimization problem, however, the methods only uses observational data and are non-trivial to extend to interventional data. In our paper, we present a method that uses continuous optimization methods that works on both observational and interventional data.

For interventional data, it is often assumed that the models have access to full intervention information, which is rare in the real world. Rothenhäusler et al. (2015) have investigated the case of additive shift interventions, while Eaton & Murphy (2007b) have examined the situation where the targets of experimental interventions are imperfect or uncertain. This is different from our setting where the intervention is unknown to start with and is assumed to arise from other agents and the environment.

Learning based methods have been proposed (Guyon, 2013; 2014; Lopez-Paz et al., 2015) and there also exist recent approaches using the generalization ability of neural networks to learn causal signals from purely observational data (Kalainathan et al., 2018; Goudet et al., 2018). Neural network methods equipped with learned masks, such as (Ivanov et al., 2018; Li et al., 2019; Yoon et al., 2018; Douglas et al., 2017), exist in the literature, but only a few (Kalainathan et al., 2018) have been adapted to causal inference. This last work is, however, tailored for causal inference on continuous variables and from observations only. Adapting it to a discrete-variable setting is made difficult by its use of a Generative Adversarial Network (GAN) Goodfellow et al. (2014) framework.

## 4 STRUCTURE DISCOVERY FROM INTERVENTIONS METHOD

***Scope of Applicability and Objective.*** The proposed method, like any structure learning algorithm, assumes the availability of a data-generating process based on ancestral sampling of a ground-truth SCM of $M$ variables, which can be queried for samples. The SCM supports applying and retracting known or unknown interventions. The method can support infinite- or finite-data as well as infinite- or finite-intervention regimes.

The objective is, then, to learn the SCM's structure from the insights that each intervention gives about cause-effect relationships between variables in the SCM.

### 4.1 PROBLEM SETTING AND ASSUMPTIONS

In this paper, we restrict the problem setting to specific, but still broad classes of SCMs and interventions. In particular, we assume that:

***Data is discrete-valued.*** The SCM's random variables are all categorical.

***Causal sufficiency.*** For every data sample, the value of all variables are available; There are no latent confounders.

***Interventions are localized.*** They affect only a single variable (but which one may not be known).

***Interventions are soft.*** An intervention does not necessarily pin its target random variable to a fixed value (though it may, as a special case). It changes the relationship of a variable with its parents.

***Interventions do not stack.*** Before a new intervention is made, the previous one is fully retracted. This stops the SCM from wandering away from its initial, observational configuration after a long series of interventions.

***No control over interventions.*** The structure learning algorithm has control neither of the target, nor the nature of the next intervention on the SCM.

For a detailed description of the interventions, refer to §A.2.

### 4.2 VARIATIONS AND PRIOR KNOWLEDGE

In the problem setting above, the ground-truth SCM is completely opaque to us. However, we consider two interesting relaxations of this formulation:

***Complete or partial graph recovery.*** We may already know the existence of certain cause-effect edges and non-edges within the ground-truth SCM. If such prior information is available, it turns a *complete graph recovery* problem into one of *partial graph recovery*. Larger SCMs can be tackled if only parts of the graph need to be recovered.

***Known or unknown interventions:*** The interventions can either be known or unknown to the learned model.

We demonstrate that the proposed method can naturally incorporate this prior information to improve its performance.

### 4.3 METHOD OVERVIEW

The proposed method is a *score-based*, *iterative*, *continuous-optimization* method consisting of three phases that flow into one other (See Figure 2). During the three-phase procedure, a structural representation of a DAG and a functional representation of a set of independent causal mechanisms are trained jointly until convergence. Because the structural and functional parameters are not independent and do influence each other, we train them in alternating phases, a form of block coordinate descent optimization.

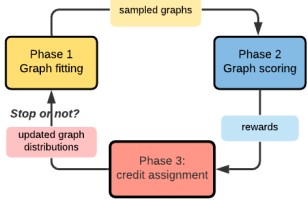

Figure 2: Workflow for our proposed method SDI. Phase 1 samples graphs under the model's current belief about the edge structure and fits parameters to observational data. Phase 2 scores a small set of graphs against interventional data and assigns rewards according to graphs' ability to predict interventions. Phase 3 uses the rewards from Phase 2 to update the beliefs about the edge structure. If the believed edge probabilities have all saturated near 0 or 1, the method has converged.

#### 4.3.1 PARAMETRIZATION

We distinguish two sets of parameters: The *structural parameters* $\gamma$ and the *functional parameters* $\theta$. Given a graph of $M$ variables, we parametrize the structure $\gamma$ as a matrix $\mathbb{R}^{M \times M}$ such that $\sigma(\gamma_{ij})$ is our belief in random variable $X_j$ being a direct cause of $X_i$, where $\sigma(x) = 1/(1 + \exp(-x))$ is the sigmoid function. The matrix $\sigma(\gamma)$ is thus a soft *adjacency matrix*.

The set of functional parameters $\theta_i$ parametrizes the conditional probability distribution of $X_i$ given its parent set $X_{\mathrm{pa}(i,C)}$, with $C \sim \mathrm{Ber}(\sigma(\gamma))$ a hypothesized configuration of the SCM's DAG.

#### 4.3.2 PHASE 1: GRAPH FITTING ON OBSERVATIONAL DATA

During Phase 1, the functional parameters $\theta$ are trained to maximize the likelihood of randomly drawn observational data under graphs randomly drawn from our current beliefs about the edge structure. We draw graph configurations $C_{ij} \sim \mathrm{Ber}(\sigma(\gamma_{ij}))$ and batches of observational data from the unintervened ground-truth SCM, then maximize the log-likelihood of the batch under that configuration using SGD. The use of graph configurations sampling from Bernoulli distributions is analogous to dropout on the inputs of the functional models (in our implementation, MLPs), giving us an ensemble of neural networks that can model the observational data.

#### 4.3.3 PHASE 2: GRAPH SCORING ON INTERVENTIONAL DATA

During Phase 2, a number of graph configurations are sampled from the current edge beliefs parametrized by $\gamma$, and scored on data samples drawn from the intervention SCM.

**Intervention applied:** At the beginning of Phase 2, an intervention is applied to the ground-truth SCM. This intervention is not under the control of the method. In our implementation, and unbeknownst to the model, the target variable is chosen uniformly randomly from all $M$ variables throughout the optimization process.

**Intervention predicted:** If the target of the intervention is not known, it is *predicted* using a simple heuristic. A small number of interventional data samples are drawn from the SCM and more graphs are sampled from our current edge beliefs. The average log-likelihood of each individual variable $X_i$ across the samples is then computed using the functional model parameters $\theta$ fine-tuned on observational data in Phase 1. The variable $X_i$ showing the greatest deterioration in log-likelihood is assumed to be the target because the observational distribution most poorly predicts that variable.

If the target of the intervention is known, then this is taken as ground-truth knowledge for the purpose of subsequent steps, and no prediction needs to be done.

**Graphs Sampled and Scored:** A new set of interventional data samples and graph configurations are now drawn from the intervention SCM and edge beliefs respectively. The log-likelihood of the data batches under the hypothesized configurations is computed, with one modification: The contribution to the total log-likelihood of a sample $X$ coming from the target (or predicted-target) intervention variable $X_i$ is masked. Because $X_i$ was intervened upon (in the manner of a Pearl do-operation, soft or hard), the values one gets for that variable should be taken as givens, not as contributors to the total log-likelihood of the sample. As well, no gradient should be allowed to propagate into the variable's learned functional parametrization $\theta_i$, because it was not actually responsible for the outcome.

**Intervention retracted:** After Phase 2, the intervention is retracted, per our modelling assumptions.

### 4.3.4 PHASE 3: CREDIT ASSIGNMENT TO STRUCTURAL PARAMETERS

During Phase 3, the scores of the interventional data batches over various graph configurations are aggregated into a gradient for the structural parameters $\gamma$. Because a discrete Bernoulli random sampling process was used to sample graph configurations under which the log-likelihoods were computed, we require a gradient estimator to propagate gradient through to the $\gamma$ structural parameters. Several alternatives exist, but we adopt for this purpose the REINFORCE-like gradient estimator $g_{ij}$ proposed by Bengio et al. (2019):

$$g_{ij} = \frac{\sum_k (\sigma(\gamma_{ij}) - c_{ij}^{(k)}) \mathcal{L}_{C,i}^{(k)}(X)}{\sum_k \mathcal{L}_{C,i}^{(k)}(X)}, \quad \forall i,j \in \{0, \ldots, M-1\} \tag{2}$$

where the $^{(k)}$ superscript indicates the values obtained for the $k$-th draw of $C$ under the current edge beliefs parametrized by $\gamma$. Therefore, $\mathcal{L}_{C,i}^{(k)}(X)$ can be read as the log-likelihood of variable $X_i$ in the data sample $X$ under the $k$'th configuration, $C^{(k)}$, drawn from our edge beliefs. Using the estimated gradient, we then update $\gamma$ with SGD, and return to Phase 1 of the continuous optimization process.

The gradient estimator $g_{ij}$ minimizes an implicit empirical risk objective with respect to $\gamma_{ij}$. When the functional and structural parameters $\theta$ and $\gamma$ are "sufficiently close" to their minima, the estimator $g_{ij}$ empirically converges quickly towards that minimum $\gamma^*$ as shown in Figure 16 of Appendix A.13.

**Acyclic Constraint:** We include a regularization term $J_{\text{DAG}}(\gamma)$ that penalizes length-2 cycles in the learned adjacency matrix $\sigma(\gamma)$, with a tunable strength $\lambda_{\text{DAG}}$. The regularization term is $J_{\text{DAG}}(\gamma) = \sum_{i \neq j} \cosh(\sigma(\gamma_{ij})\sigma(\gamma_{ji})), \quad \forall i,j \in \{0, \ldots, M-1\}$ and is derived from Zheng et al. (2018). The details of the derivation are in the Appendix. We explore several different values of $\lambda_{\text{DAG}}$ and their effects in our experimental setup. Suppression of longer-length cycles was not found to be worthwhile for the increased computational expense.

## 5 EXPERIMENTAL SETUP AND RESULTS

We first evaluate the proposed method on a synthetic dataset where we have control over the number of variables and causal edges in the ground-truth SCM. This allows us to analyze the performance of the proposed method under various conditions. We then evaluate the proposed method on real-world datasets from the BnLearn dataset repository. We also consider the two variations of §4.2: Recovering only part of the graph (when the rest is known), and exploiting knowledge of the intervention target.

The summary of our findings is: 1) We show strong results for graph recovery for all synthetic graphs in comparisons with other baselines, measured by Hamming distance. 2) The proposed method achieves high accuracy on partial graph recovery for large, real-world graphs. 3) The proposed method's intervention target prediction heuristic closes the gap between the known- and unknown-target intervention scenarios. 4) The proposed method generalizes well to unseen interventions. 5) The proposed method's time-to-solution scaling appears to be driven by the number of edges in the groundtruth graph moreso than the number of variables.

## 5.1 MODEL DESCRIPTION

**Learner model.** Without loss of generality, we let $\theta_i = \{\texttt{W0}_i, \texttt{B0}_i, \texttt{W1}_i, \texttt{B1}_i\}$ define a stack of $M$ one-hidden-layer MLPs, one for each random variable $X_i$. A more appropriate model, such as a CNN, can be chosen using domain-specific knowledge; the primary advantage of using MLPs is that the hypothesized DAG configurations $c_{ij}$ can be readily used to mask the inputs of MLP $i$, as shown in Figure 3.

To force the structural equation $f_i$ corresponding to $X_i$ to rely exclusively on its direct ancestor set $\text{pa}(i, C)$ under hypothesis adjacency matrix $C$ (See Eqn. 1), the one-hot input vector $X_j$ for variable $X_i$'s MLP is masked by the Boolean element $c_{ij}$. An example of the multi-MLP architecture with $M$=4 categorical variables of $N$=3 categories is shown in Figure 3. For more details, refer to Appendix A.4.

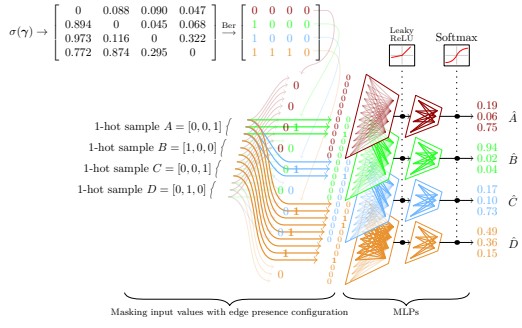

Figure 3: MLP Model Architecture for an $M = 4$, $N = 3$ SCM. The model computes the conditional probabilities of $A$, $B$, $C$, $D$ given their parents using a stack of four independent MLPs. The MLP input layer uses an adjacency matrix sampled from $\text{Ber}(\sigma(\gamma))$ as an input mask to force the model to make use only of parent nodes to predict their child node.

**Ground-truth model.** Ground-truth SCM models are parametrized either as CPTs with parameters from BnLearn (in the case of real-world graphs), or as a second stack of MLPs similar to the learner model, with randomly-initialized functional parameters $\theta_{\text{GT}}$ and the desired adjacency matrix $\gamma_{\text{GT}}$.

**Interventions.** In all experiments, at most one (soft) intervention is concurrently performed. To simulate a soft intervention on variable $X_i$, we reinitialize its ground-truth conditional distribution's MLP parameters or CPT table randomly, while leaving the other variables untouched. For more details about the interventions, please refer to Appendix A.2.

## 5.2 SYNTHETIC DATASETS EXPERIMENTS

We first evaluate the model's performance on several randomly-initialized SCMs with specific, representative graph structures. Since the number of possible DAGs grows super-exponentially with the number of variables, for $M$=4 up to 13 a selection of

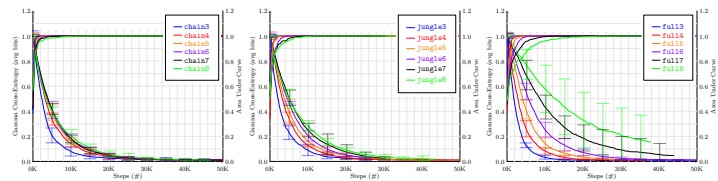

Figure 4: Cross entropy (CE) and Area-Under-Curve (AUC/AUROC) for edge probabilities of learned graph against ground-truth for synthetic SCMs. Error bars represent $\pm 1\sigma$ over PRNG seeds 1-5. **Left to right**: chainM,jungleM,fullM,$M = 3 \ldots 8$ ($9 \ldots 13$ in Appendix A.6.1). Graphs (3-8 variables) all learn perfectly with AUROC reaching 1.0. However, denser graphs (fullM) take longer to converge.

representative and edge-case DAGs are chosen. chainM and fullM ($M$=3-13) are the minimally-and maximally-connected $M$-variable DAGs, while treeM and jungleM are tree-like intermediate graphs. colliderM is the $(M-1) \rightarrow 1$ collider graph. The details of the setup is in Appendix A.6.

**Results.** The model can recover most synthetic DAGs with high accuracy, as measured by Structural Hamming Distance (SHD) between learned and ground-truth DAGs. Table 1 shows our proposed method outperforming all other baseline methods, and learns all graphs perfectly for 3 to 13 variables (excepting full). For DAGs ranging from 3 to 8 variables, the AUROCs all eventually reach 1.0 (indicating perfect classification into edge/non-edge; Refer to Figure 4). For both large ($M > 10$)

Table 1: **Baseline comparisons:** Structural Hamming Distance (SHD) (lower is better) for learned and ground-truth edges on various graphs from both synthetic and real datasets, compared to (Peters et al., 2016), (Heinze-Deml et al., 2018b), (Eaton & Murphy, 2007b), (Yu et al., 2019) and (Zheng et al., 2018). The proposed method (Structural Discovery from Interventions (SDI)) is run on random seeds $1 - 5$ and we pick the worst performing model out of the random seeds in the table. OOM: out of memory. Our proposed method correctly recovers the true causal graph, with the exception of Sachs and full13, and it significantly outperforms all other baseline methods. Proposed method as well as all the baselines uses similar amount of data.

| Method | Asia | Sachs | collider | chain | jungle | collider | full |
| --- | --- | --- | --- | --- | --- | --- | --- |
| $M$ | 8 | 11 | 8 | 13 | 13 | 13 | 13 |
| **Zheng et al. (2018)** | 14 | 22 | 18 | 39 | 22 | 24 | 71 |
| **Yu et al. (2019)** | 10 | 19 | 7 | 14 | 16 | 12 | 77 |
| **Heinze-Deml et al. (2018b)** | 8 | 17 | 7 | 12 | 12 | 7 | 28 |
| **Peters et al. (2016)** | 5 | 17 | 2 | 2 | 8 | 2 | 16 |
| **Eaton & Murphy (2007a)** | 0 | OOM | 7 | OOM | OOM | OOM | OOM |
| **Proposed Method** (SDI) | 0 | 6 | 0 | 0 | 0 | 0 | 7 |

and dense DAGs (e.g. `full13`) the model begins encountering difficulties, as shown in Table 1 and Appendix §A.6.1.

Small graphs ($M < 10$) are less sensitive than larger ones to our hyperparameters, notably the sparsity and acyclic regularization (§4.3.4) terms. In §A.5, we perform an analysis of these hyperparameters.

## 5.3 REAL-WORLD DATASETS: BNLEARN

The Bayesian Network Repository is a collection of commonly-used causal Bayesian networks from the literature, suitable for Bayesian and causal learning benchmarks. We evaluate the proposed method on the Earthquake (Korb & Nicholson, 2010), Cancer (Korb & Nicholson, 2010), Asia (Lauritzen & Spiegelhalter, 1988) and Sachs (Sachs et al., 2005) datasets ($M =$5, 5, 8 and 11-variables respectively, maximum in-degree 3) in the BnLearn dataset repository.

**Results.** As shown in Table 1, the proposed method perfectly recovers the DAG of Asia, while making a small number of errors (SHD=6) for Sachs (11-variables). It thus significantly outperforms all other baselines models. Figures 8 & 9 visualize what the model has learned at several stages of learning. Results for Cancer and Asia can be found in the appendices, Figure 17 and 18.

## 5.4 COMPARISONS WITH OTHER METHODS

As shown in Table 1, we compared the proposed SDI method to ICP ((Peters et al., 2016)), non-linear ICP ((Heinze-Deml et al., 2018b)), and (Eaton & Murphy, 2007b; Zheng et al., 2018; Yu et al., 2019) on Asia (Lauritzen & Spiegelhalter, 1988), Sachs (Sachs et al., 2005) and representative synthetic graphs. Eaton & Murphy (2007b) handles uncertain interventions and Peters et al. (2016), Heinze-Deml et al. (2018b) handles unknown interventions. However, neither attempts to predict the intervention. As shown in Table 1, we significantly outperform ICP, non-linear ICP, and the methods in (Yu et al., 2019) and (Zheng et al., 2018). Furthermore, Eaton & Murphy (2007b) runs out of memory for graphs larger than $M = 10$ because modelling of uncertain interventions is done using "shadow" random variables (as suggested by the authors), and thus recovering the DAG internally requires solving a $d = 2M$-variable problem. Their method's extremely poor time- and space-scaling of $O(d2^d)$ makes it unusable beyond $d > 20$.

For SDIs, we threshold our edge beliefs at $\sigma(\gamma) = 0.5$ to derive a graph, but the continued decrease of the cross-entropy loss (Figure 4) hints at SDI's convergence onto the correct causal model. Please refer to Appendix §A.8 for full details and results.

## 5.5 GENERALIZATION TO PREVIOUSLY UNSEEN INTERVENTIONS

It is often argued that machine learning approaches based purely on capturing joint distributions do not necessarily yield models that generalize to unseen experiments, since they do not explicitly model changes through interventions. By way of contrast,

Table 2: **Evaluating the consequences of a previously unseen intervention:** (test log-likelihood under intervention)

| | fork3 | chain3 | confounder3 | collider3 |
| --- | --- | --- | --- | --- |
| **Baseline** | -0.5036 | -0.4562 | -0.3628 | -0.5082 |
| **SDI** | -0.4502 | -0.3801 | -0.2819 | -0.4677 |

causal models use the concept of interventions to explicitly model changing environments and thus hold the promise of robustness under distributional shifts (Pearl, 2009; Schölkopf et al., 2012; Peters et al., 2017). To test the robustness of causal modelling to previously unseen interventions (new values for an intervened variable), we evaluate a well-trained causal model against a variant, non-causal model trained with $c_{ij} = 1, \ i \neq j$. An intervention is performed on the ground-truth SCM, fresh interventional data is drawn from it, and the models, with knowledge of the intervention target, are asked to predict the other variables given their parents. The average log-likelihoods of the data under both models are computed and contrasted. The intervention variable's contribution to the log-likelihood is masked. For all 3-variable graphs (`chain3`, `fork3`, `collider3`, `confounder3`), the causal model attributes higher log-likelihood to the intervention distribution's samples than the non-causal variant, thereby demonstrating causal models' superior generalization ability in transfer tasks. Table 2 collects these results.

### 5.6 VARIANT: PREDICTING INTERVENTIONS

In Phase 2 (§4.3.3), we use a simple heuristic to predict the intervention target variable. Experiments show that this heuristic functions well in practice, yielding correct predictions far more often than by chance alone (Table 3). Guessing the intervention variable randomly, or not guessing it at all, leads to a significant drop in the model performance, even for 3-variable graphs (Fig. 11 Left). Training SDI with intervention prediction closely tracks training with leaked knowledge of the ground-truth intervention on larger, 7-variable graphs (Fig. 11 Right).

Table 3: **Intervention Prediction Accuracy:** (identify on which variable the intervention took place)

| 3 variables | 4 variables | 5 variables | 8 variables |
|---|---|---|---|
| 95 % | 93 % | 85 % | 71 % |

### 5.7 VARIANT: PARTIAL GRAPH RECOVERY

Instead of learning causal structures *de novo*, we may have partial information about the ground-truth SCM and may only need to fill in missing information (§4.2). An example is protein structure discovery in biology, where some causal relationships have been definitely established and others remain open hypotheses. This is an easier task compared to full graph recovery, since the model only has to search for missing edges. We evaluate the proposed method on Barley (Kristensen & Rasmussen, 2002) ($M = 48$) and Alarm (Beinlich et al., 1989) ($M = 37$) from the BnLearn repository. The model is asked to predict 50 edges from Barley and 40 edges from Alarm. The model reached $\geq 90\%$ accuracy on both datasets, as shown in Table 4.

Table 4: **Partial Graph Recovery** on Alarm (Beinlich et al., 1989) and Barley (Kristensen & Rasmussen, 2002). The model is asked to predict 50 edges in Barley and 40 edges in Alarm. The accuracy is measured in Structural Hamming Distance (SHD). SDI achieved over 90% accuracy on both graphs.

| Graph | Alarm | Barley |
|---|---|---|
| Number of variables | 37 | 48 |
| Total Edges | 46 | 84 |
| Edges to recover | 40 | 50 |
| Recovered Edges | 37 | 45 |
| Errors (in SHD) | 3 | 5 |

### 5.8 ABLATION AND ANALYSIS

As shown in Figure 12, larger graphs (such as $M > 6$) and denser graphs (such as `full8`) are progressively more difficult to learn. For denser graphs, the learned models have higher sample complexity, higher variance and slightly worse results. Refer to Appendix §A.9 for complete results on all graphs. **Hyperparameters.** Hyperparameters for all experiments were kept identical unless otherwise stated. We study the effect of DAG and sparsity penalties in the following paragraph. For more details, please refer to Appendix §A.5 .

**Importance of regularization.** Valid configurations $C$ for a causal model are expected to be a) sparse and b) acyclic. To promote such solutions, we use DAG and sparsity regularization with tunable hyperparameters. We set the DAG penalty to $0.5$ and sparsity penalty to $0.1$. We run ablation studies on different values of the regularizers and study their effect. We find that smaller graphs are less sensitive to different values of regularizer than larger graphs. For details, refer to Appendix §A.12.

**Importance of dropout.** To train functional parameter for an observational distribution, sampling adjacency matrices is required. We "drop out" each edge (with a probability of $\sigma(\gamma)$) in our experiments during functional parameter training of the conditional distributions of the SCM. Please refer to Appendix §A.14 for a more detailed analysis.

## 6 CONCLUSION

In this work, we introduced an experimentally successful method (SDI) for causal structure discovery using continuous optimization, combining information from both observational and interventional data. We show in experiments that it can recover true causal structure, that it generalizes well to unseen interventions, that it compares very well against start-of-the-art causal discovery methods on real world datasets, and that it scales even better on problems where only part of the graph is known.

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

# Appendix

## Table of Contents

## A    Annexes

### A.1    Training Algorithm

Algorithm 1 shows the pseudocode of the method described in §4. Typical values for the loop trip counts are found in §A.11.

### A.2    Preliminaries

**Interventions.**    In a purely-observational setting, it is known that causal graphs can be distinguished only up to a Markov equivalence class. In order to identify the true causal graph intervention data is needed (Eberhardt et al., 2012). Several types of common *interventions* may be available (Eaton & Murphy, 2007b). These are: *No intervention:* only observational data is obtained from the ground truth causal model. *Hard/perfect:* the value of a single or several variables is fixed and then ancestral sampling is performed on the other variables. *Soft/imperfect:* the conditional distribution of the variable on which the intervention is performed is changed. *Uncertain:* the learner is not sure of which variable exactly the intervention affected directly. Here we make use of soft interventions for several reasons: First, they include hard interventions as a limiting case and hence are more general. Second, in many real-world scenarios, it is more difficult to perform a hard intervention compared to a soft one. We also deal with a special case of uncertain interventions, where the variable selected for intervention is random and unknown. We call these *unidentified* or *unknown* interventions.

**Intervention setup.**    For our experiments, the groundtruth models of the synthetic datasets are modeled by neural networks as described in section A.6. Each neural network models the relationship of the causal parents and a variable. We perform our intervention by first randomly selecting which variable to intervene on, then soft-intervening on it. The selected variable is sampled from a uniform distribution. The soft intervention is a reinitialization of its neural network's parameters.

**Causal sufficiency.**    The inability to distinguish which causal graph, within a Markov equivalence class, is the correct one in the purely-observational setting is called the *identifiability problem*. In our setting, all variables are observed (there are no latent confounders) and all interventions are random and independent. Hence, within our setting, if the interventions are known, then the true causal

---

**Algorithm 1** Training Algorithm

---

1: **procedure** TRAINING(SCM Ground-Truth Entailed Distribution $D$, with $M$ nodes and $N$ categories)
2:     **Let** $i$ an index from 0 to $M - 1$

3:     **for** $I$ iterations, or until convergence, **do**
4:         **if** $I$ % reinitialization_period == 0 **then**
5:             $D \leftarrow$ reinitialize($D$)

6:         **for** $F$ functional parameter training steps **do**                    ▷ **Phase 1**
7:             $X \sim D$
8:             $C \sim \text{Ber}(\sigma(\gamma))$
9:             $L = -\log P(X|C\,;\theta)$
10:            $\theta_{t+1} \leftarrow \text{Adam}(\theta_t, \nabla_\theta L)$

11:        **for** $Q$ interventions **do**
                                                                ▷ **Phase 2**
12:            I_N $\leftarrow$ randint(0, $M - 1$)                 ▷ Uniform selection of target
13:            $D_{\text{int}} := D$ with intervention on node I_N         ▷ Apply intervention

14:            **if** predicting intervention **then**                ▷ Phase 2 Prediction
15:                $L_i \leftarrow 0 \quad \forall i$
16:                **for** $N_P$ prediction steps **do**
17:                    $X \sim D_{\text{int}}$
18:                    **for** $C_P$ configurations **do**
19:                        $C \sim \text{Ber}(\sigma(\gamma))$
20:                        $L_i \leftarrow L_i - \log P_i(X|C_i;\theta_{\text{slow}}) \,\forall i$
21:                I_N $\leftarrow$ argmax($L_i$)

22:            gammagrads, logregrets = [], []                    ▷ Phase 2 Scoring
23:            **for** $N_S$ scoring steps **do**
24:                $X \sim D_{\text{int}}$
25:                gammagrad, logregret = 0, 0
26:                **for** $C_S$ configurations **do**
27:                    $C \sim \text{Ber}(\sigma(\gamma))$
28:                    $L_i = -\log P_i(X|C_i;\theta_{\text{slow}}) \quad \forall i$
29:                    gammagrad += $\sigma(\gamma) - C$              ▷ Collect $\sigma(\gamma) - C$ for Equation 2
30:                    logregret += $\sum\limits_{i \neq \text{I\_N}} L_i$         ▷ Collect $\mathcal{L}_{C,i}^{(k)}(X)$ for Equation 2

31:                gammagrads.append(gammagrad)
32:                logregrets.append(logregret)

                                                                ▷ **Phase 3**
33:            $g_{ij} = \dfrac{\sum_k (\sigma(\gamma_{ij}) - c_{ij}^{(k)})\mathcal{L}_{C,i}^{(k)}(X)}{\sum_k \mathcal{L}_{C,i}^{(k)}(X)}$     ▷ Gradient Estimator, Equation 2
34:            $g \leftarrow g + \nabla_\gamma (\lambda_{\text{sparse}} L_{\text{sparse}}(\gamma) + \lambda_{\text{DAG}} L_{\text{DAG}}(\gamma))$     ▷ Regularizers
35:            $\gamma_{t+1} \leftarrow \text{Adam}(\gamma_t, g)$

---

graph is always identifiable in principle (Eberhardt et al., 2012; Heinze-Deml et al., 2018a). We also consider here situations where a single variable is randomly selected and intervened upon with a soft or imprecise intervention, its identity is unknown and must be inferred. In this case, there is no theoretical guarantee that the causal graph is identifiable. However, there is existing work Peters et al. (2016) that handles this scenario and the proposed method is also proven to work empirically.

**Faithfulness.** It is possible for causally-related variables to be probabilistically independent purely by happenstance, such as when causal effects along multiple paths cancel out. This is called *unfaithfulness*. We assume that *faithfulness* holds, since the $\gamma$ gradient estimate is extracted from shifts in probability distributions. However, because of the "soft" nature of our interventions and their infinite variety, it would be exceedingly unlikely for cancellation-related unfaithfulness to persist throughout the causal-learning procedure.

### A.3 EXPERIMENTAL SETUP

For all datasets, the weight parameters for the learned model is initialized randomly. In order to not bias the structural parameters, all $\sigma(\gamma)$ are initialized to $0.5$ in the beginning of training. Details of hyperparameters of the learner model are described in Section A.5. The experimental setup for the groundtruth model for the synthetic data can be found in Section A.6 and the details for the real world data are described in Section A.7.

### A.4 MODEL SETUP

As discussed in section 4, we model the $M$ variables in the graph using $M$ independent MLPs, each possesses an input layer of $M \times N$ neurons (for $M$ one-hot vectors of length $N$ each), a single hidden layer chosen arbitrarily to have $\max(4M, 4N)$ neurons with a LeakyReLU activation of slope $0.1$, and a linear output layer of $N$ neurons representing the unnormalized log-probabilities of each category (a softmax then recovers the conditional probabilities from these logits). To force $f_i$ to rely exclusively on the direct ancestor set $pa(i, C)$ under adjacency matrix $C$ (See Eqn. 2), the one-hot input vector $X_j$ for variable $X_i$'s MLP is masked by the Boolean element $c_{ij}$. The functional parameters of the MLP are the set $\theta = \{\texttt{W0}_{ihjn}, \texttt{B0}_{ih}, \texttt{W1}_{inh}, \texttt{B1}_{in}\}$. An example of the multi-MLP architecture with $M$=3 categorical variables of $N$=2 categories is shown in Figure 3.

### A.5 HYPERPARAMETERS

**Learner model.** All experiments on the synthetic graphs of size 3-8 use the same hyperparameters. Both the functional and structural parameters are optimized using the Adam optimizer Kingma & Ba (2014). We use a learning rate of $5e - 2$ with alpha of $0.9$ for the functional parameters, and we use a learning rate of $5e - 3$ with alpha of $0.1$ for the structural parameters. We perform 5 runs of each experiment with random seeds $1 - 5$ and error bars are plotted for various graphs from size 3 to 8 in Figure 4. We use a batch size of 256. The L1 norm regularizer is set to $0.1$ and the $DAG$ regularizer is set to $0.5$ for all experiments. For each $\gamma$ update step, we sample 25 structural configurations from the current $\gamma$. In all experiments, we use 100 batches from the interventional distribution to predict the intervened node.

### A.6 SYNTHETIC DATA

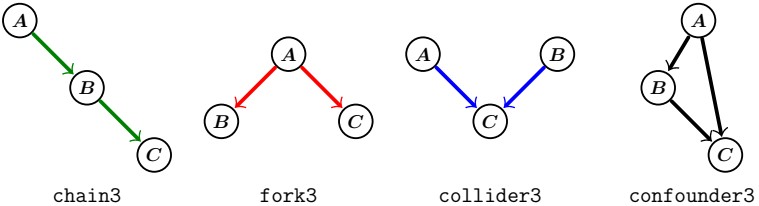

Figure 5: Every possible 3-variable connected DAG.

**Synthetic datasets.** The synthetic datasets in the paper are modeled by neural networks. All neural networks are 2 layered feed forward neural networks (MLPs) with Leaky ReLU activations between layers. The parameters of the neural network are initialized orthogonally within the range of $(-2.5, 2.5)$. This range was selected such that they output a non-trivial distribution. The biases are initialized uniformly between $(-1.1, 1.1)$.

SCM with $n$ variables are modeled by $n$ feedforward neural networks (MLPs) as described in §5.1. We assume an acyclic causal graph so that we may easily sample from them. Hence, given any pair of random variables $A$ and $B$, either $A \rightarrow B$, $B \rightarrow A$ or $A$ and $B$ are independent.

The MLP representing the ground-truth SCM has its weights $\theta$ initialized use orthogonal initialization with gain $2.5$ and the biases are initialized using a uniform initialization between $-1.1$ and $1.1$, which was empirically found to yield "interesting" yet learnable random SCMs.

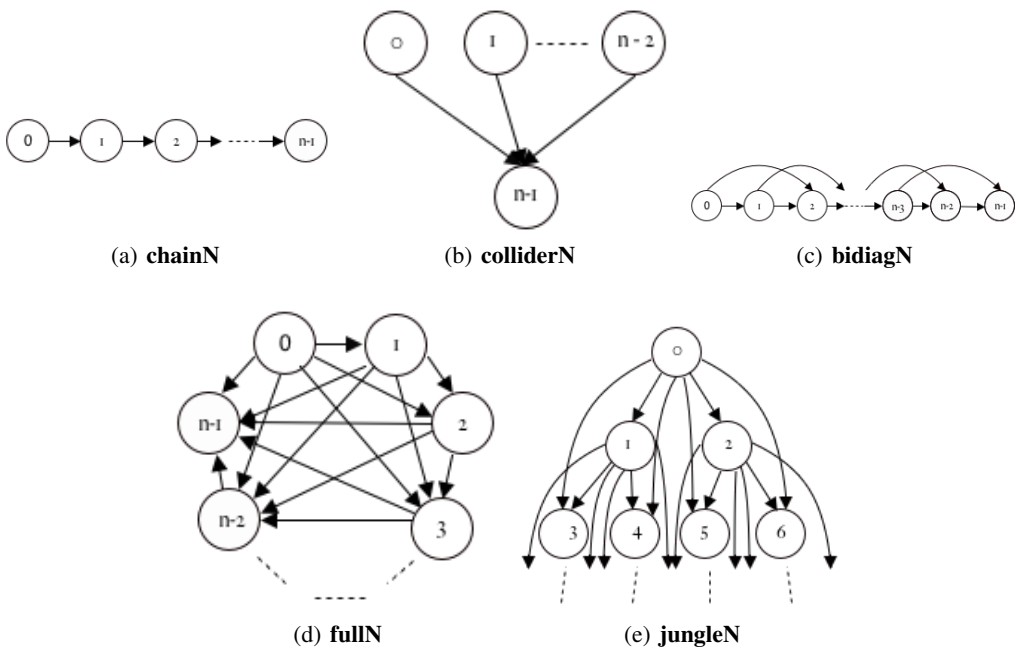

Figure 6: Figures for various synthetic graphs. chain, collider, bidiagonal, full and jungle graph.

We study a variety of SCMs with different ground-truth edge structures $\gamma$. Our selection of synthetic graphs explores various extremes in the space of DAGs, stress-testing SDI. The chain graphs are the sparsest connected graphs possible, and are relatively easy to learn. The bidiag graphs are extensions of chain where there are 2-hops as well as single hops between nodes, doubling the number of edges and creating a meshed chain of forks and colliders. The jungle graphs are binary-tree-like graphs, but with each node connected directly to its grandparent in the tree as well. Half the nodes in a jungle graph are leaves, and the out-degree is up to 6. The collider graphs deliberately collide independent $M - 1$ ancestors into the last node; They stress maximum in-degree. Lastly, the full graphs are the maximally dense DAGs. All nodes are direct parents of all nodes below them in the topological order. The maximum in- and out-degree are both $M - 1$. These graphs are depicted in Figure 6.

### A.6.1 Synthetic data results

The model can recover correctly all synthetic graphs with 10 variables or less, as shown in Figure 10 and Table 1. For graphs larger than 10 variables, the model found it more challenging to recover the denser graphs (e.g. fullM), as shown in Table 1. Plots of the training curves showing average cross entropy (CE) and Area-Under-Curve(AUC/AUCROC) for edge probabilities of the learned graph against the ground-truth graph for synthetic SCMs with 3-13 variables are available in Figure 10.

### A.7 BnLearn data repository

The repo contains many datasets with various sizes and structures modeling different variables. We evaluate the proposed method on 3 of the datasets in the repo, namely the Earthquake (Korb & Nicholson, 2010), Cancer (Korb & Nicholson, 2010) and Asia (Lauritzen & Spiegelhalter, 1988) datasets. The ground-truth model structure for the Cancer (Korb & Nicholson, 2010) and Earthquake (Korb & Nicholson, 2010) datasets are shown in Figure 7. Note that even though the structure for the two datasets seems to be the same, the conditional probability tables (CPTs) for these datasets are very different and hence results in different structured causal models (SCMs) for each.

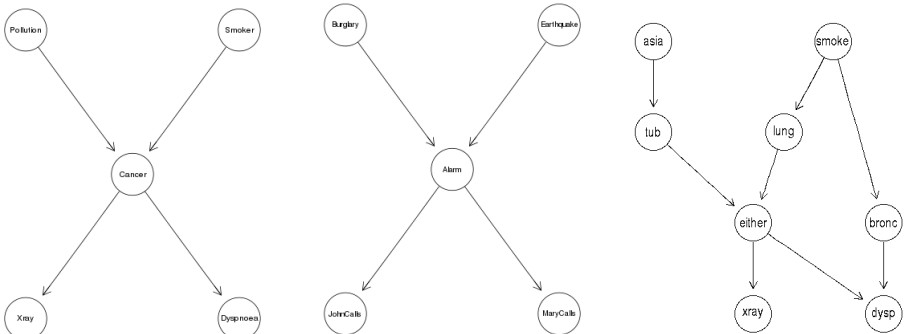

Figure 7: Left to right: Ground Truth SCM for Cancer, Groundtruth SCM for Earthquake, Groundtruth SCM for Asia.

| Method | Asia | chain8 | jungle8 | collider7 | collider8 | full8 |
|---|---|---|---|---|---|---|
| **(Zheng et al., 2018)** | 14 | 24 | 14 | 11 | 18 | 21 |
| **(Yu et al., 2019)** | 10 | 7 | 12 | 6 | 7 | 25 |
| **(Heinze-Deml et al., 2018b)** | 8 | 7 | 12 | 6 | 7 | 28 |
| **(Peters et al., 2016)** | 5 | 3 | 8 | 4 | 2 | 16 |
| **(Eaton & Murphy, 2007a)** | 0 | 0 | 0 | 7 | 7 | 1 |
| SDIs | 0 | 0 | 0 | 0 | 0 | 0 |

Table 5: **Baseline comparisons:** Hamming distance (lower is better) for learned and ground-truth edges on various graphs from both synthetic and real datasets, compared to (Peters et al., 2016), (Heinze-Deml et al., 2018b), (Eaton & Murphy, 2007b), (Yu et al., 2019) and (Zheng et al., 2018). The proposed SDI is run on random seeds $1 - 5$ and we pick the worst performing model out of the random seeds in the table.

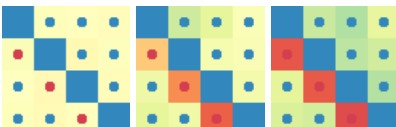  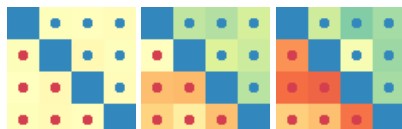

Figure 8: Learned edges at three different stages of training. **Left**: `chain4` (chain graph with 4 variables). **Right**: `full4` (tournament graph with 4 variables).

## A.8 COMPARISONS TO OTHER METHODS

As described in section 5.4, we compare to 5 other methods. The full comparison between SDIs and other methods on various graphs can be found in Table 1.

One of these methods, DAG-GNN Yu et al. (2019), outputs 3 graphs based on different criteria: best mean square error (MSE), best negative loglikelihood (NLL) and best evidence lower bound (ELBO). We report performance of all outputs of DAG-GNN Yu et al. (2019) in Table 6, and the best one is selected for Table 1.

## A.9 SPARSITY OF GROUND-TRUTH GRAPH

We evaluated the performance of SDI on graphs of various size and sparsity to better understand the performance of the model. We evaluated the proposed model on 4 representative types of graphs in increasing order of density. They are the `chain`, `jungle`, `bidiag` and `full` graphs. As shown in the results in figure 12, for graphs of size 5 or smaller, there is almost no difference in the final results in terms of variance and sample complexity. However, as the graphs gets larger (than 6), the denser graphs (`full` graphs) gets progressively more difficult to learn compared to the sparser graphs (`chain`, `jungle` and `bidiag`). The models learned for denser graphs have higher complexity, higher variance and slightly worse results.

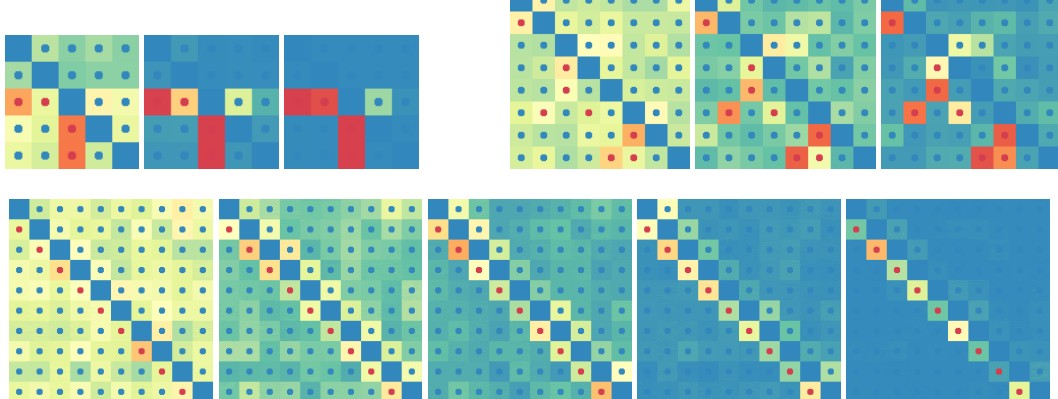

Figure 9: **Top Left:** Earthquake: Learned edges at three different stages of training. **Top Right:** Asia: Learned edges at three different stages of training. **Bottom:** `chain10` at different stages of training, clearly displaying Markov-equivalence of causal and anti-causal chain. Training resolves in causal direction after further training.

|  | **SDI** | Best MSE | Best NLL | Best Elbo |
|---|---|---|---|---|
| `Asia` | 0 | 10 | 10 | 13 |
| `chain8` | 0 | 7 | 7 | 7 |
| `jungle8` | 0 | 12 | 12 | 13 |
| `collider7` | 0 | 6 | 6 | 6 |
| `collider8` | 0 | 8 | 8 | 7 |
| `full8` | 0 | 27 | 25 | 27 |

Table 6: **Baseline comparisons:** Hamming distance (lower is better) for learned and ground-truth edges on `Asia` and various synthetic graphs. compared to DAG-GNN Yu et al. (2019). DAG-GNN outputs 3 graphs according to different criterion. We show results on all outputs in this table and we show the best performing result in Table 1.

## A.10 PREDICTING INTERVENTIONS

In Phase 2, we score graph configurations based on how well they fit the interventional data. We find that it is necessary to avoid disturbing the learned parameters of intervened variables, and to ignore its contribution to the total negative log-likelihood of the sample. Intuitively, this is because, having been intervened upon, that variable should be taken as a given. It should especially not be interpreted as a poorly-learned variable requiring a tuning of its functional parameters, because those functional parameters were not responsible for the value of that variable; The extrinsic intervention was.

Since an intervened variable is likely to be unusually poorly predicted, we heuristically determine that the most poorly predicted variable is the intervention variable. We then zero out its contribution to the log-likelihood of the sample and block gradient into its functional parameters.

Figure 11 illustrates the necessity of this process. When using the prediction heuristic, the training curve closely tracks training with ground-truth knowledge of the identity of the intervention. If no prediction is made, or a random prediction is made, training proceeds much more slowly, or fails entirely.

## A.11 SAMPLE COMPLEXITY

Our method is heavily reliant on sampling of configurations and data in Phases 1 and 2. We present here the breakdown of the sample complexity. Let

- $I$ be the number of iterations of the method, *(typical: 500-2000)*
- $B$ the number of samples per batch, *(typical: 256)*
- $F$ the number of functional parameter training iterations in Phase 1, *(typical: 10000)*
- $Q$ the number of interventions performed in Phase 2, *(typical: 100)*
- $N_P$ the number of data batches for prediction, *(typical: 100)*

| | Chain | | | | | | Jungle | | | | | | Full | | | | | |
|---|---|---|---|---|---|---|---|---|---|---|---|---|---|---|---|---|---|---|
| | 3 | 4 | 5 | 6 | 7 | 8 | 3 | 4 | 5 | 6 | 7 | 8 | 3 | 4 | 5 | 6 | 7 | 8 |
| **ldag=0.5, lsparse=0.1** | 0 | 0 | 0 | 0 | 0 | 0 | 0 | 0 | 0 | 0 | 0 | 0 | 0 | 0 | 0 | 0 | 0 | 0 |
| *ldag=0.5, lsparse=0.* | 0 | 0 | 0 | 0 | 0 | 0 | 0 | 0 | 0 | 0 | 0 | 0 | 0 | 0 | 0 | 0 | 0 | 0 |
| *ldag=0., lsparse=0.1* | 0 | 0 | 0 | **1** | 0 | 0 | 0 | 0 | 0 | 0 | **1** | **3** | **1** | 0 | 0 | 0 | **1** | **6** |

Table 7: **Regularizer:** SDI performance measured by Hamming distance to the ground-truth graph. Comparisons are between SDIs with different regularizer settings for different graphs. Our default setting is ldag = 0.5, lsparse = 0.1, with ldag the DAG regularization strength and lsparse the sparsity regularization strength. As shown in the table, SDIs is not very sensitive to different regularizer settings. Tasks with non-zero Hamming distance (errors) are in bold.

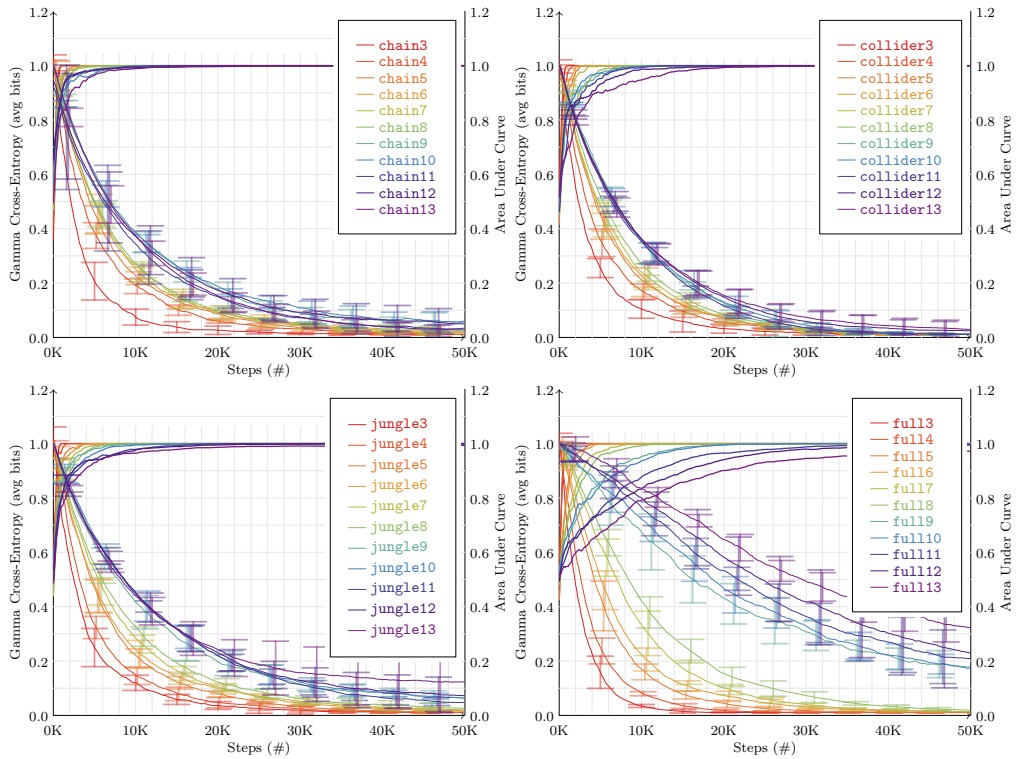

Figure 10: Cross entropy (CE) and Area-Under-Curve (AUC/AUROC) for edge probabilities of learned graph against ground-truth for synthetic SCMs. Error bars represent $\pm 1\sigma$ over PRNG seeds 1-5. **Left to right, up to down**: chainM,jungleM,fullM,$M = 3\ldots8$ ($9\ldots13$ in Appendix A.6.1). Graphs (3-13 variables) all learn perfectly with AUROC reaching 1.0. However, denser graphs (fullM) take longer to converge.

- $C_P$ the number of graph configurations drawn per prediction data batch, *(typical: 10)*
- $N_S$ the number of data batches for scoring, *(typical: 10)*
- $C_S$ the number of graph configurations drawn per scoring data batch. *(typical: 20-30)*

Then the total number of interventions performed, and configurations and samples drawn, over an entire run are:

$$\text{Interventions} = IQ = \gamma \text{ updates} \tag{3}$$

$$\text{Samples} = I(\underbrace{F}_{\text{Phase 1}} + \underbrace{Q(N_P + N_S)}_{\text{Phase 2}})B \tag{4}$$

$$\text{Configurations} = I(\underbrace{F}_{\text{Phase 1}} + \underbrace{Q(C_P N_P + C_S N_S)}_{\text{Phase 2}}) \tag{5}$$

Because of the multiplicative effect of these factors, the number of data samples required can quickly spiral out of control. For typical values, as many as $500 \times 10000 \times 256 = 1.28\text{e}9$ observational and $500 \times 100 \times (100 + 10) \times 256 = 1.408\text{e}9$ interventional samples are required. To alleviate this problem slightly, we limit the number of samples generated for each intervention; This limit is usually 500-2000.

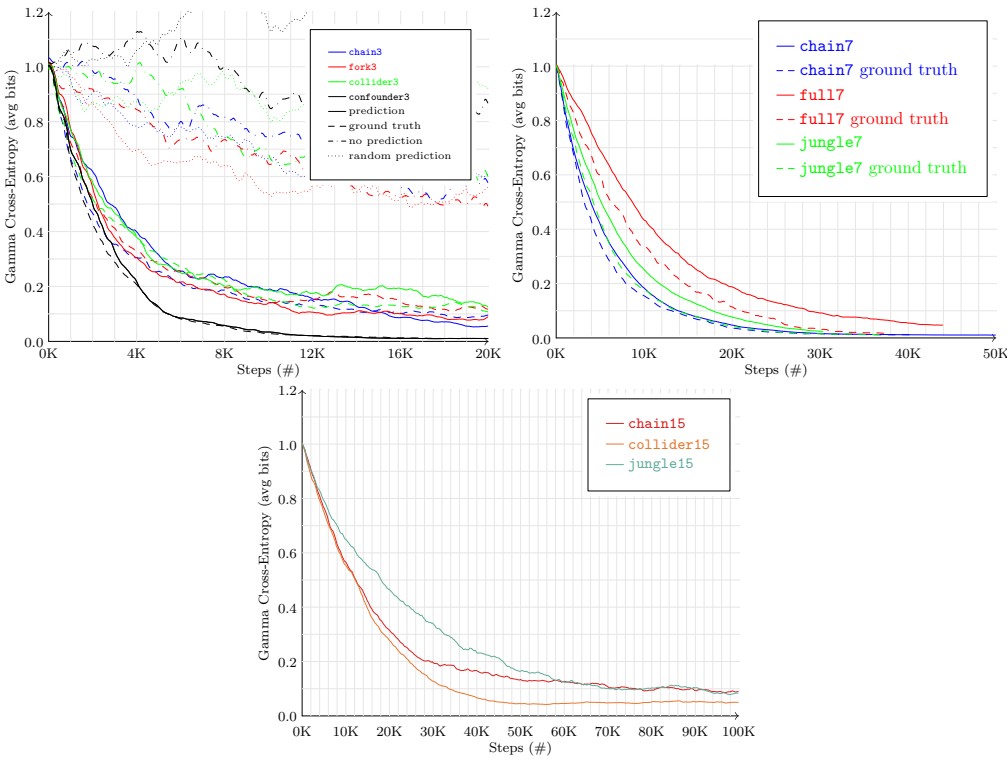

Figure 11: **Ablation Study of Intervention Prediction** Cross-entropy loss over time on multiple graphs and intervention prediction modes. **Above Left**: All 3-variable graphs. Solid/dashed lines: Ground-truth & Prediction strategies. Dotted lines: Random- & No-Prediction strategies. Training with prediction closely tracks ground-truth. **Above Right**: Comparison for 7-variable graphs, ground-truth against prediction strategy. Training with prediction still closely tracks ground-truth at larger scales. **Below**: Performance on 15-variable graphs with known intervention targets.

## A.12 EFFECT OF REGULARIZATION

**Importance of sparsity regularizer.** We use a $L1$ regularizer on the structure parameters $\gamma$ to encourage a sparse representation of edges in the causal graph. In order to better understand the effect of the $L1$ regularizer, we conducted ablation studies on the $L1$ regularizer. It seems that the regularizer has an small effect on rate of converges and that the model converges faster with the regularizer, This is shown in Figure 13. However, this does not seem to affect the final value the model converges to, as is shown in Table 7.

**Importance of DAG regularizer.** We use an acyclic regularizer to discourage length-2 cycles in the learned model. We found that for small models ($\leq 5$ variables), the acyclic regularizer helps with faster convergence, without improving significantly the final cross-entropy. This is illustrated for the 3-variable graphs in Figure 14. However, for graphs larger than 5 variables, the acyclic regularizer starts playing an important role in encouraging the model to learn the correct structure. This is shown in the ablation study in Table 7.

## A.13 NEAR-OPTIMUM PERFORMANCE OF GRADIENT ESTIMATOR

The gradient estimator $g_{ij}$ we use to minimize the empirical risk w.r.t. the structural parameters $\gamma$, defined in Eq. 2 is adapted from Bengio et al. (2019). We verify that the estimator samples the correct gradient by an experiment that tests convergence near the optimum.

To do this, we pre-initialize the structural and functional parameters near the global minimum, and verify that $\gamma$ converges. Specifically, the ground-truth functional parameters $\theta$ are copied and disturbed by a small Gaussian noise, while the ground-truth structural parameters $\gamma$ are copied, but the confidences in an edge or non-edge are set to 88% and 12% rather than 100% and 0%. The experiment is then expected to quickly converge to the global minimum.

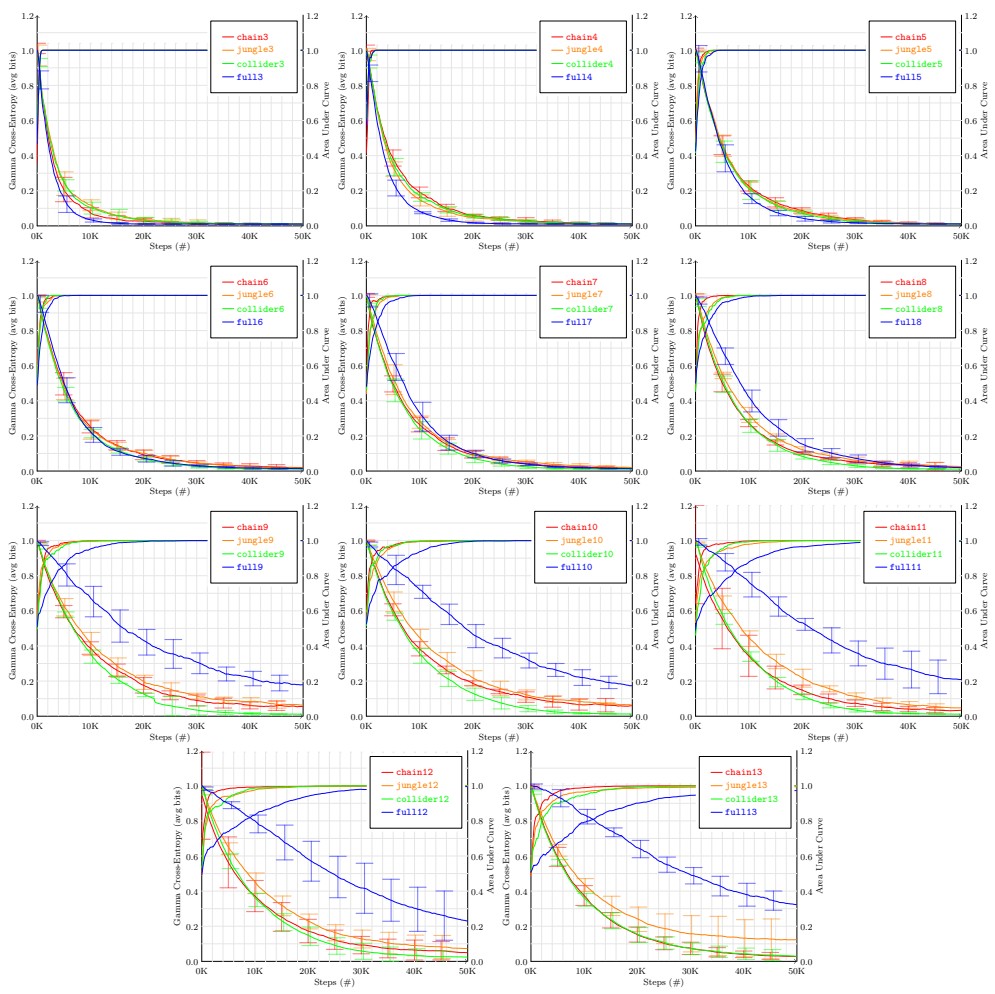

Figure 12: **Left to right, top to bottom** Average cross-entropy loss of edge beliefs $\sigma(\gamma)$ and Area-Under-Curve throughout training for the synthetic graphs `chainN`, `jungleN`, `colliderN` and `fullN`, $N$=3-13, grouped by graph size. Error bars represent $\pm 1\sigma$ over PRNG seeds 1-5.

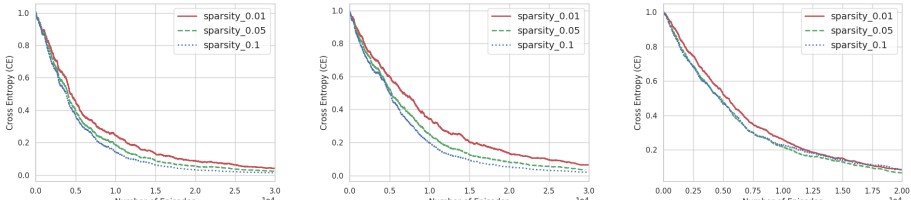

Figure 13: **Effect of sparsity ($lsparse$) regularizer :** On 5 variable, 6 variable and 8 variable Nodes

As shown in Figure 16, the gradient estimator correctly enables Stochastic Gradient Descent towards the minimum, for the `chain` and `jungle` graphs of size 15, 20 and 25. The average cross-entropy rapidly approaches its floor of 0.01, a consequence of our clamping of all $\gamma_{ij}$ to the range $\pm 5$ (equivalently, clamping $\sigma(\gamma_{ij})$ to the range $[0.0067, 0.9933]$).

## A.14    IMPORTANCE OF DROPOUT

To train the functional parameters on an observational distribution, one would need sampling adjacency matrices. One may be tempted to make these "complete directed graph" (all-ones except for a zero diagonal), to give the MLP maximum freedom to learn any potential causal relations itself. We

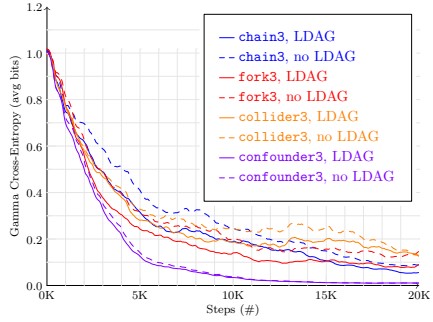

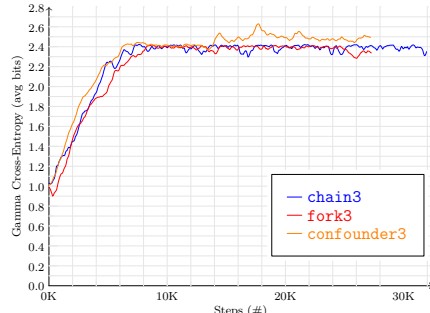

Figure 14: Ablations study results on all possible 3 variable graphs. Graphs show the cross-entropy loss on learned vs ground-truth edges over training time. Comparisons of model trained with and without DAG regularizer ($L_{DAG}$), showing that DAG regularizer helps convergence.

Figure 15: Edge CE loss for 3-variable graphs with no dropout when training functional parameters, showing the importance of this dropout.

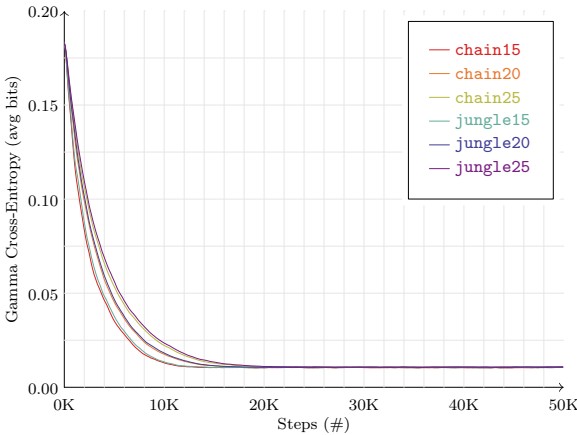

Figure 16: **Near-optima performance of gradient estimator:** Performance on chain and jungle, $M = 15, 20, 25$, and initialized from near the global optimum. Illustrates correctness and rapid convergence using the gradient estimator in Eq. 2 near the optimum.

demonstrate that functional parameter training cannot be carried out this way, and that it is necessary to "drop out" each edge (with probability of the current $\gamma$ value in our experiments) during pre-training of the conditional distributions of the SCM. We attempt to recover the previously-recoverable graphs chain3, fork3 and confounder3 without dropout, but fail to do so, as shown in Figure 15.

|  | SDI | Eaton & Murphy (2007b) |
|---|---|---|
| Asia | 0 | 0 |
| chain8 | 0 | 0 |
| jungle8 | 0 | 0 |
| collider7 | 0 | 7 |
| collider8 | 0.0 | 7 |
| full8 | 0.0 | 1 |

Table 8: **Comparisons:** Structured hamming distance (SHD) on learned and ground-truth edges on asia and various synthetic graphs. Eaton & Murphy (2007b) can not scale to larger variables graphs as shown in Table 1, hence, we compare to the largest graph that (Eaton & Murphy, 2007b) can scale up to. SDI is compared to (Eaton & Murphy, 2007b) for collider7, collider8 and full8, (Eaton & Murphy, 2007a) asserts with 100% confidence a no-edge where there is one (false negative). For comparisons with all other methods 1.

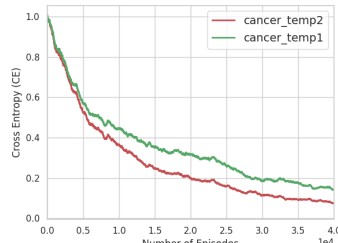

Figure 17: Cross-entropy for edge probability between learned and ground-truth SCM for Cancer at varying temperatures.

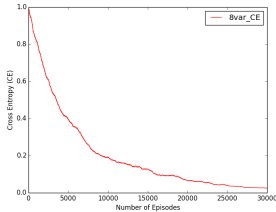

Figure 18: Cross-entropy for edge probability between learned and ground-truth SCM. **Left**: The Earthquake dataset with 6 variables. **Right**: The Asia dataset with 8 variables

