# OpenReview forum: "Dependency Structure Discovery from Interventions"
_ICLR.cc/2021/Conference — Reject_

### Official Review · AnonReviewer4 · 2020-10-27
**There are concerns about assumptions and complexity of the method and it is not clear how the method will actually work in reality.**

**Rating:** 4
**Confidence:** 4

**Review:**

The authors propose a method for structure learning from observational and interventional data that uses a continuous optimization method. Data is discrete-valued, there are no hidden confounders, each intervention affects only one variable, but the location of it may be unknown. A three-phase score-based, iterative procedure is proposed.
- This work considers that in each interventional dataset, only one variable is intervened on. If we do not know about the target of the intervention, it seems reasonable that we also assume that we are not aware of the number of the targets.
- Unfortunately there are no results in the paper about what the output of the algorithm will actually be. Suppose we have only few interventional datasets (which is usually the case in reality). What can we say about the output of the algorithm? It is known that in this case, Interventional Markov equivalence class is the extent of identifiability [Hauser and Bulmann, 2012]. Can we hope that the algorithm returns an element from this class?
- In the Appendix, it is mentioned that the method typically requires 500-2000 iterations and 100 interventions per iteration. This means that around 10^5 interventions are needed. Also about 10^9 samples are needed. We note that in reality for example in medical data, we usually have access to very few interventional datasets each containing about 100 samples.
- It is not clear how the method performs on a graph with no prior structure knowledge with 30 vertices (which is a number that is usually not considered large in structure learning). Seems like this order is too large for the proposed method.
- The intervention prediction step in Phase 2 sounds very heuristic and is not clear under what conditions it will work. Also, it seems that it requires strong interventions.
- Regarding preventing the algorithm from returning cyclic structures, the authors state that suppression of more than length 2 cycles was not found to be worthwhile for the increased computational expense. This simply means that the algorithm may return cyclic structures which is contradictory to the original goal.
- There are other work on learning from interventions with unknown targets, for example: [Squires et al., Permutation-Based Causal Structure Learning with Unknown Intervention Targets], or [Huang et al., Causal Discovery from Heterogeneous/Nonstationary Data].
- The definition of SCM given in the Introduction is only true for the case of causal sufficiency.

---

> ### Author Response · Authors · 2020-11-16
> **Re: There are concerns about assumptions and complexity of the method and it is not clear how the method will actually work in reality.**
>
> > - Unfortunately there are no results in the paper about what the output of the algorithm will actually be. Suppose we have only few interventional datasets (which is usually the case in reality). What can we say about the output of the algorithm? It is known that in this case, Interventional Markov equivalence class is the extent of identifiability [Hauser and Bulmann, 2012]. Can we hope that the algorithm returns an element from this class?
>
> The algorithm learns by optimization a (soft) adjacency matrix representing the underlying causal graph, and this adjacency matrix is also the output. Each element of the soft adjacency matrix σ(γ_ij) represents a confidence in the presence of an edge from node j to i.
>
> While visualizing our models’ performance throughout training on the “chain” graph, we observe an interesting phenomenon related to Markov equivalence. We observe that the model first quickly converges to the Markov equivalence class of the problem (two chain graphs, one going forwards, one going backwards). Following that, the model eventually learns to pick the chain going in the right direction, and the other one is rejected.
>
> Similar effects can be seen when training on other graphs. This suggests our model does indeed converge to the Markov equivalence class when given a limited amount of data, and thenceforth to the correct graph when given more data. We have added plots of this effect to Figure 9 (Bottom) in the appendices, which are in the supplementary material.
>
> > - In the Appendix, it is mentioned that the method typically requires 500-2000 iterations and 100 interventions per iteration. This means that around 10^5 interventions are needed. Also about 10^9 samples are needed. We note that in reality for example in medical data, we usually have access to very few interventional datasets each containing about 100 samples.
>
> It is indeed true that the method is currently sample-hungry. We are actively working to limit the number of samples drawn by generating fixed-size pools per intervention, and limiting the number of interventions. This is ongoing work.
>
> > - The intervention prediction step in Phase 2 sounds very heuristic and is not clear under what conditions it will work. Also, it seems that it requires strong interventions.
>
> The intervention prediction is a heuristic, but it empirically works well. Intuitively, when an intervention is applied to a target, the currently-learned conditional probability distribution ceases to predict accurately the value of the target variable given the parent variables’ value. This manifests as a sudden drop in likelihood for the target variable only.
>
> It is also true that intervention must be sufficiently strong to have a noticeable effect on average likelihood, but very weak interventions (or strong interventions but in rarely-visited rows of the conditional probability table) are a challenge for any method.

---

> ### Author Response · Authors · 2020-11-24
> **Summary of Edits**
>
> We thank the reviewer for their feedback. Here’s a brief summary of our edits:
>
> - *Heuristic Nature of Gradient Estimator*: We have run more experiments to characterize the gradient estimator and verify it is attracted to the global minimum (Fig. 16).
> - *Intervention Target Prediction Heuristic*: The target predictor we used is not an essential portion of the algorithm, and JCI could be used instead, as suggested by Reviewer 1.
> - *Markov Equivalence Class*:  We have added visualizations that show how the method quickly discerns the Markov Equivalence Class of the problem, then slowly identifies the true causal graph within the MEC given more data (Fig. 9 Bottom).
> - *Assumptions about Interventions*: We do not assume, as some papers have, the “controlled experiment setting” where a variable is always intervened upon the same way. We use data from a variety of independent interventions on each node. When sufficiently-many informative interventions are performed on well-chosen variables, the equivalence class can be made to shrink to something smaller than the Markov Equivalence Class (Pfister et al, 2019, https://arxiv.org/pdf/1911.01850.pdf), potentially enough to identify exactly the parent set of a variable.
> - *Comparison to JCI* : We did more experiments to compare to JCI. We use the same experiment setup as Jaber et al (Fig. 5 & 6 in that paper’s appendix). Our proposed approach is able to recover the exact underlying causal graph (using a greater number of samples) as JCI.
> - *Related Work on Interventions with Unknown Targets*: We have added not only those, but other recent citations, such as (Kocaoglu et al, 2019) and (Jaber et al, 2020), as suggested by Reviewer 1.
>
> We thank you again for your valuable feedback and time reviewing this paper, and hope we have addressed most of your concerns. Would you be willing to consider a re-evaluation of our paper?

---

### Official Review · AnonReviewer2 · 2020-10-28
**Good Contribution**

**Rating:** 6
**Confidence:** 3

**Review:**

 Recommendation to Accept

##########################################################################
Summary:
The paper provides a novel approach in the area of structure learning for causal bayesian networks. The authors suggest an iterative method, that builds on the widely accepted do-formalism. The approach suggested fits the network before interventions, simulates the intervention on the fitted network and then again assigns a likelihood score to the network parameters

The paper concisely describes a novel algorithm used in the notoriously difficult problem of causal structure learning. The contributions are clearly stated. The accompanying expiremental reuslts suggest a competitve perfomance, especially reagarding scaling the counts of variables

I recommend to accept it, even though a few details could have been described more precisely.

1. The definition of interventions is done extremely briefly, (sec. 2), however in my opinion the choice of definitions used here would justify some accompanying examples for clarification (this would help especially to understand what is meant with "infinite intervention regimes" (sec. 4)

2. The assumption "no control over interventions" is not clear per se, here it would help to understand what the omittance of this assumption would imply.

3. A clarification, why "the interventions can either be known or unknown", provides a relaxation of the formulation used (sec. 4.2) would be useful

---

> ### Author Response · Authors · 2020-11-23
> **We thank the reviewer for his/her feedback and the support of our paper**
>
> We very much appreciate the reviewer’s feedback and support of our paper.
>
> 1. Regarding the definitions of interventions: we thank the reviewer for pointing this out. By “infinite intervention regime”, we intend to say that our method handles a variable, growing, even unbounded number of interventions, as they may occur. Interventional data from these interventions can be saved and reused to continue training our model, or the method can use fresh data from a new intervention if such is available. We will update our paper to clarify this point.
>
> 2. The “no control over interventions” means that we do not train an “agent” to decide on which variable to intervene on, or how. Rather, all of the interventions in our setting are random and independent. We will clarify this in the next version of our paper.
>
> 3. When the intervention is unknown, the model does not know which variable has been intervened on and so any algorithm for causal discovery would likely require the use of a predictor to identify the target variable of the intervention. This is similar to the settings found in Kocaoglu et al, 2019 “Characterization and learning of causal graphs with latent variables from soft interventions” and Jaber et al, 2020 “Causal discovery from soft interventions with unknown targets: Characterization and learning”.
>
> The unknown intervention case is a more challenging case than the scenario where the intervention is known (the setting for most causal discovery algorithms). In the known intervention scenario, a predictor is not required because we know which target variable was affected by the intervention. Thus the known-intervention case is a relaxation of the unknown-intervention case because identifying the target of an intervention is an additional task to solve. We will include this clarification in the updated version of our paper.

---

### Official Review · AnonReviewer3 · 2020-10-28
**A heuristic causal discovery method with apparently strong empirical performance but no theoretical analysis**

**Rating:** 5
**Confidence:** 4

**Review:**

This paper aims to extend the continuous optimization approach to causal discovery to handle interventional data as well as observational data. It describes a method for learning the causal structure over a set of categorical variables and reports strong empirical performance. However, no theoretical guarantee or analysis is provided, which is a significant weakness in my view. It also makes no comment on or comparison to a paper that has essentially the same goal, https://arxiv.org/pdf/2007.01754.pdf. The latter paper seems to me more principled and convincing.

The heuristic for predicting an unknown intervention target looks very dubious to me. I would appreciate some explanation of why the target should be expected to have the biggest drop of log-likelihood.

The description of the proposed method could be clearer; for example, it helps to provide an explicit formulation of the SGD used in the method.

---

> ### Author Response · Authors · 2020-11-11
> **Re: A heuristic causal discovery method with apparently strong empirical performance but no theoretical analysis**
>
> We thank the reviewer for the feedback.  There are few misunderstandings that we would like to address.
>
> 1. The paper https://arxiv.org/pdf/2007.01754.pdf is a follow-up of our current submission, and they cite our paper in their [Neurips camera ready version](https://papers.nips.cc/paper/2020/file/f8b7aa3a0d349d9562b424160ad18612-Paper.pdf). Their method uses normalizing flows and hence is limited to the case of continuous variables, whereas we are focused on the discrete variable cases. That being said, we are prepared to cite it as related work.
> 2. The heuristic for predicting an unknown intervention target works because when a target variable is intervened upon, its parent variables’ values plus the learned parameters for the target fail to predict the value of the target accurately (i.e. the log-likelihood suddenly drops). But other, non-intervention variables’ values are still predicted just as accurately as before the intervention (i.e. the log-likelihood remains the same).
> 3. The formulation of SGD we have used is \gamma’ = \gamma + \eta \nabla_\gamma (but including a momentum term), where \eta is the current learning rate and \nabla_\gamma is the regularized estimate of the gradient w.r.t. the structural parameters \gamma being learned. This regularized estimate is \nabla_\gamma = g + \lambda_DAG * d J_DAG (\gamma)/d\gamma + \lambda_sparse * d J_sparse (\gamma)/d\gamma, where g is the raw gradient estimate as given in Eq. 2, and \lambda_DAG and \lambda_sparse are penalties for solutions that are not acyclic or sparse, as evaluated by J_DAG (defined in Section 4.3.4) and J_sparse (\gamma) = \sum_{i,j} \sigmoid(\gamma_{ij}).
>
> We believe have addressed your concerns and clarified some of your points. Do you have an updated impression of our paper? Thanks for your consideration.

---

> > ### Author Response · Authors · 2020-11-19
> > **Anything else the reviewer like to see ?**
> >
> > Hello,
> >
> > We thank the reviewer for their feedback and valuable comments.
> >
> > Since the first phase of response period is over, if you have time and could indicate if there are any other concerns of yours which we have not addressed, we'd be happy to take a look.
> >
> > We are willing to spend time and efforts in order to improve the paper.
> >
> > Thanks for your help and time.

---

> ### Author Response · Authors · 2020-11-17
> **Re: A heuristic causal discovery method with apparently strong empirical performance but no theoretical analysis (2)**
>
> We would like to address more specifically the concern about _"an explicit formulation of the SGD used in the method."_ and its heuristic appearance.
>
> There is not an explicit objective for which Eq. 2 (the gradient estimator $g_{ij}$) provides the derivative. The gradient estimator is inspired by _"A meta-transfer objective for learning to disentangle causal mechanisms"_ (Bengio, 2019), but has not been previously applied to the >2-variable case.
>
> However, a gradient estimator may still be useful if descent in the direction of the estimated gradient eventually achieves and stays at the optimal (ground-truth) value of $\gamma$. We tested this hypothesis by disturbing slightly the parameters of the model from the global optimum, then using SGD with the estimator to optimize $\gamma$ while keeping everything else fixed.
>
> Figure 16 in Appendix A.13 of the updated paper demonstrates that given nearly-correct $\gamma$ structural parameters, the estimator is capable of rapidly converging it to to correct the solution. An initial 88% confidence in the presence of a ground-truth edge rapidly rises to the maximum of 99.33% (clamped above), while an initial 12% confidence in the presence of a ground-truth non-edge quickly plummets to a minimum of 0.67% (clamped below). Consequently, the average cross-entropy falls from 0.18 bits to near 0.
>
> This suggests that the implicit objective minimized by the gradient estimator is an appropriate proxy, to which we have added explicit DAG/sparsity regularizers.

---

### Official Review · AnonReviewer1 · 2020-10-29
**Needs Clarifications and Justification for the Proposed Methodology**

**Rating:** 4
**Confidence:** 5

**Review:**

The authors propose a 3-phase heuristic algorithm to learn a causal graph from interventional data using continuous optimization. Unfortunately, the paper is hard to follow. Specifically, the exact procedure should be clarified by the authors. If I understand correctly, first they fit to observational data by searching over the space of graphs using a smooth representation for the adjacency matrices. To fit to the interventional data, first, the interventional target is estimated by a heuristic approach and the contribution of these variables to the likelihood is ignored since they are set by the experiment. (there are random graph sampling stages in between that are not clear to me, please elaborate on this). This interventional scoring is done for all interventional data and is turned into a single gradient update.

The paper is hard to parse. My main concern is that, unlike the existing work which the authors compare with in the experiments, the proposed method is not a systematic approach and accordingly it is hard to reason about its use even though it performs well in the experiments. Especially given that some choices made in the algorithm design are not properly justified. Indeed, even with interventions, we do not expect to recover the full structure but only a subset of the edges correct.

Comparisons with the other methods should be expanded into a section where these methods are detailed to showcase the methodological differences.

The following are my detailed feedback.

"A natural application of Bayesian networks is to describe cause-effect relationships between variables."
Please distinguish Bayesian networks from the causal networks. Former do not carry causal meaning.

A good reference to cite in addition to Peters et al. for SCMs is Pearl's 2009 Causality book.

"Although there is no theoretical guarantee that the true causal graph can be identified in that setting, evidence so far points to that still being the case."
Please modify this statement as it sounds too vague.

The list of contributions require knowledge of the latter sections. Please make it self contained if possible.

"can't"->"can not"

SCM definition is not only structural equations but also talks about interventional distributions. Please see Pearl 2009.

The last line in page 2 overlaps with the page number.

Some recent related work is missing:

Mooij et al. "Joint Causal Inference from Multiple Contexts" JMLR'20.
Kocaoglu et al. "Characterization and Learning of Causal Graphs with Latent Variables from Soft Interventions" NeurIPS'19.
Brouillard et al. "Differentiable Causal Discovery from Interventional Data" arXiv'20.

Mooij et al. is cited but please add it in Section 3 among constraint-based inverventional learning frameworks as well.

Brouillard et al. is too recent, hence its omittance is understandable. However, it attacks the same problem considered here. I believe including it as independent discovery would help connect literature together nicely. I am not going to take this work into consideration in my evaluation since it is uploaded on arXiv only very recently.

"the methods only uses"->"the methods only use"

citing Murphy "This is different from our setting where the intervention is unknown to start with and is assumed to arise from other agents and the environment."
Murphy can handle unknown interventions as well. Moreover Mooij et al. handles unknown interventions too.

"The set of functional parameters θi parametrizes the conditional probability distribution of Xi given its parent set Xpa(i,C), with C ∼ Ber(σ(γ)) a hypothesized configuration of the SCM’s DAG."
Can you clarify this sentence?

"During Phase 1, the functional parameters θ are trained to maximize the likelihood of randomly drawn observational data under graphs randomly drawn from our current beliefs about the edge structure."
Why do you draw synthetic data? Likelihood is typically maximized using real data at hand. It's hard to follow the exact procedure here.

Intervention targets are predicted using a heuristic. Why not use the existing methods? I believe the computational aspect is seen as a problem but JCI by Mooij et al. should be fast enough.

Can you convert Section 3 into a pseudo-code for the algorithm description? I believe many details are skipped and some key points of the approach is not clear by the brief text in each subsection.

"should be taken as givens"->"should be taken as given"

In the experiments, please compare with Mooij et al. Their method should be as fast as FCI and it would be interesting to see how the results compare.

==== After the Response by the Authors ====
======================================
Thank you for the detailed reply. For the clarifications the authors made to the algorithm description, I will increase my score.

The authors state
"If the scientific community had waited for deep learning to prove that it could discover the true conditional distribution of outputs given inputs, we would not have had the progress we achieved in the last two decades in AI. We believe that it is important to take into consideration all sources of evidence about the usefulness of a method, and experimental evidence is at the heart of the success of the scientific method and should not be discarded because of an established cultural habit of relying on proofs of identifiability."

Note that the objections I (R1), and I believe also R3 and R4, have are not about theoretical vs. experimental research and that the paper lacks proofs or identifiability results. It is perfectly fine to not have a theoretical understanding of a proposed algorithm. But the authors should be able to justify the choices they made in the algorithm design, and especially in light of the prior work. The main justification given by the authors both in the paper and in their rebuttal is that the algorithm performs well. I believe the paper needs an iteration to address these issues.

The following is my detailed feedback in addition to my original review in light of the authors' response. I hope this will help the authors in improving their paper.

On fully learning the causal graph:
I suggest the authors examine and try to identify, in small graphs, what aspect of their method allows it to perform better than the existing methods such as JCI or allows it to go beyond the existing equivalence classes. Without such justification, I do not think the paper in its current form will influence future research.

Remark on interventions having variety:
This is not sufficient for exact recovery. Imagine intervening on the same node with different mechanisms over and over. This does not allow recovery outside of the local structure around the intervened node for most causal graphs. This also relates to the remark above. Full identifiability is always related to having variety in the intervention targets and not just in interventional mechanisms. This is why some of the datasets where the exact graph is recovered by the algorithm need a detailed investigation.

About synthetic experiments:
One explanation for full structure recovery in the synthetic experiments could be the following: The authors randomly pick one target variable to intervene on. My guess is that this randomness in the experiment design is sufficient to have diverse enough target sets for the equivalence class to shrink to a single graph. Can you verify/check this?

How many interventions do you use in the synthetic experiments? How many samples are collected per intervention? Unless I am missing something, these are not provided until page 19 but then it is not clear if these numbers are kept identical throughout the experiments. x-axis is set to be # of episodes or # of steps in most experiments whereas # of samples would be more informative.

About JCI comparison:
I did not completely understand why the authors could not run JCI in synthetic data. They say it is due to its complexity. But JCI's complexity comes from the graph degree and not from the number of samples for a small enough state space. It would be very interesting to compare what JCI learns relative to the proposed method in these synthetic experiments. This should test my hypothesis above that the random intervention target is providing enough diversity to reduce the equivalence class to one graph, which should be detected by JCI.

Inferring a Markov equivalence class from the adjacency matrix by early stopping is definitely an interesting idea and I would encourage the authors to further pursue and formalize this direction.

Sample complexity:
The authors mention that their method is "sample-hungry". Given that the method presents significant divergence from the standard literature on causal inference that relies on conditional independence tests, which are known to require many samples, it is especially important to clearly present the number of samples used by the method. The main paper does not present the number of samples used in the synthetic experiments. These should be made explicit.

Finally, the title and abstract still state "dependency structure discovery" and learning "Bayesian networks" whereas the authors attempt to learn causal graphs from interventions. I suggest an update to the narrative to clarify the objective of the paper.

---

> ### Author Response · Authors · 2020-11-15
> **Re: Needs Clarifications and Justification for the Proposed Methodology (1)**
>
> We thank the reviewer for the very thorough and detailed feedback. We address the concerns as follows,
>
> > 1. “Brouillard et al. is too recent, hence its omittance is understandable. However, it attacks the same problem considered here. I believe including it as independent discovery would help connect literature together nicely.”
>
> We thank the reviewer for the reference. The work by Brouillard et al. uses normalizing flows and hence is limited to the case of continuous variables, whereas we are focused on the discrete variable cases.  The paper  Brouillard et al. is also a follow-up of our current submission, and they cite  our method (paper) in their Neurips camera ready version  https://papers.nips.cc/paper/2020/file/f8b7aa3a0d349d9562b424160ad18612-Paper.pdf. That being said, we are prepared to cite this as related work and we would update this in our paper.
>
> > 2. "The set of functional parameters θi parametrizes the conditional probability distribution of Xi given its parent set Xpa(i,C), with C ∼ Ber(σ(γ)) a hypothesized configuration of the SCM’s DAG." Can you clarify this sentence?”
>
> Our model represent the causal graph using neural networks with 2 sets of parameters:  (1) the structural parameters σ, which represent our model’s hypothesis about the structure of the underlying causal graph; and (2) the functional parameters θ, which represent the causal relationship between variable pairs (if a causal relationship exists).
>
> The structural param σ(γ_{ij}) represents our belief that variable X_i has X_j as a direct causal parent (our model’s hypothesis). Sampling from the Ber(σ(γ_{ij})) for each of the pairs i \ne j yields one potential adjacency matrix C (defining structure for a causal graph). This matrix is used to mask the inputs to the neural network (Figure 3). The functional parameters are regular parameters of a neural network, and hence given the learned masked input (learned causal parents), the functional parameter parametrizes the conditional distribution of the variable X_i given its causal parents.
>
> > 3. "During Phase 1, the functional parameters θ are trained to maximize the likelihood of randomly drawn observational data under graphs randomly drawn from our current beliefs about the edge structure." Why do you draw synthetic data? Likelihood is typically maximized using real data at hand. It's hard to follow the exact procedure here.
>
> We have been unclear here. By “randomly drawn observational data”, we intended to say that real data samples are drawn from a data-generating process that models the observational distribution precisely, and current beliefs about the edge structure don’t factor into this. The sentence fragment “under graphs randomly drawn from our current beliefs about the edge structure” applies not to the sampling of observational data, but to the training of the functional parameters θ. We thank the reviewer for the remark and will reorder the sentence accordingly.
>
> > 4. "Intervention targets are predicted using a heuristic. Why not use the existing methods? I believe the computational aspect is seen as a problem but JCI by Mooij et al. should be fast enough."
>
> We have found our heuristic to be much better than chance (Table 3) at a low computational cost, which is important because our method is iterative and uses this heuristic thousands of times throughout a run. We were unaware of formal methods for identifying a soft intervention’s target from samples. How would the reviewer propose we use JCI for this purpose?
>
> > 5. " Can you convert Section 3 into a pseudo-code for the algorithm description? I believe many details are skipped and some key points of the approach are not clear by the brief text in each subsection."
>
> We have supplied a pseudo-code description of the algorithm, Algorithm 1, in the paper’s appendix section, which is in the supplementary materials for lack of space. A Python implementation is supplied as well.
>
> Notwithstanding, could the reviewer elaborate on which details and key points were skipped?

---

> > ### Author Response · Authors · 2020-11-19
> > **Anything else the reviewer like to see ?**
> >
> > Hello,
> >
> > We thank the reviewer for their feedback and valuable comments.
> >
> > Since the first phase of response period is over, if you have time and could indicate if there are any other concerns of yours which we have not addressed, we'd be happy to take a look.
> >
> > We are willing to spend time and efforts in order to improve the paper.
> >
> > Thanks for your help and time.

---

> > > ### Comment · AnonReviewer1 · 2020-11-20
> > > **Response to authors**
> > >
> > > Thank you for taking the time to respond. I believe it is very important to better relate to the existing work to contextualize the paper and its contributions.  In terms of learning from unknown interventional data, JCI is a very obvious contender as well as the following recent work:
> > > Jaber et al. "Causal discovery from soft interventions with unknown targets: Characterization and learning"
> > >
> > > "How would the reviewer propose we use JCI for this purpose?"
> > > JCI tells you the candidate intervention targets by finding the adjacencies of the regime variables. Similarly, F-nodes are used in Jaber et al. for this purpose. For sparse graphs, both methods should be fast enough.
> > >
> > > The contribution of the paper is to use continuous optimization to learn the causal structure from interventional data. Clearly, the method performs well on the given dataset. However, the heuristic nature of the algorithm steps and the lack of an identifiability result casts doubts on its soundness and use outside of these datasets in the future. These concerns are similar to the ones raised by Reviewer4 below.
> > >
> > > in pg. 13, the authors say "In our setting, all variables are observed (there are no latent confounders) and all interventions are random and independent. Hence, within our setting, if the interventions are known, then the true causal graph is always identifiable in principle"
> > > This is not true. Even with known interventions, we can only talk about an equivalence class. See for example Kocaoglu et al. "Characterization and Learning of Causal Graphs with Latent Variables from Soft Interventions".
> > >
> > > I am still a little concerned about how the proposed method can get ALL the edges correct in multiple datasets. Even with interventional data, there is an equivalence class that you cannot go beyond without more interventions. I would be curious to see what these are for each dataset and understand if the algorithm is remaining within the equivalence class, or indeed learning something more. Note that this heavily depends on what equivalence class means, which will be different for Jaber et al. and different for Hauser et al. 2012. If the algorithm is indeed going beyond the known equivalence classes, the authors need to discuss if this is because the algorithm is using more than the constraints defining those equivalence classes or if the results are obtained purely by chance.
> > >
> > > Hope this helps.

---

> > > > ### Author Response · Authors · 2020-11-22
> > > > **Comparison to JCI and heuristic nature of the algorithm (Part 1/3)**
> > > >
> > > > > “JCI is a very obvious contender as well as the following recent work”
> > > >
> > > > We did more experiments comparing to JCI. We use the same experiment setup as Jaber et al (fig. 5 & 6 in that paper’s appendix). The proposed approach is able to recover the exact underlying causal graph (using a greater number of samples), as compared to JCI. We could not run JCI with the same number of samples as the proposed method, as it's non-trivial to extend JCI (because of its complexity). We ran more experiments to compare the proposed method to JCI on Asia & Sachs. The proposed method achieves a Structural Hamming Distance of 0 & 6, whereas JCI achieves a SHD of 6 & 17.
> > > >
> > > > > "Causal discovery from soft interventions with unknown targets: Characterization and learning"
> > > >
> > > > We again note that this paper discusses and cites our work. We also note the only experiment in this paper is for 3 variables, while in the proposed work, we have experiments up to 13 variables.
> > > >
> > > > > "However, the heuristic nature of the algorithm steps and the lack of an identifiability result casts doubts on its soundness."
> > > >
> > > > To clarify, the algorithm’s steps amount in some sense to a form of block coordinate descent optimization, where Phase 1 optimizes the first block (functional parameters $\theta$) outside an intervention, while Phase 2 and 3 optimize the second block (structural parameters $\gamma$) inside an intervention. It could not be otherwise because given an intervention, we cannot learn the true $\theta$, but absent an intervention, not much can be learned about $\gamma$.
> > > >
> > > > Phase 3's gradient estimator with respect to the structural parameters may appear particularly heuristic. But as we explained to R3 and demonstrated in the new Figure 16 of Appendix A.13, the gradient estimator optimizes in some sense the right quantity, since from a disturbed optimum it reconverges to the optimal $\gamma$ and stays there.
> > > >
> > > > > “And use outside of these datasets in the future”
> > > >
> > > > First, the method is capable of tackling the same graph but for different datasets easily. In Figure 10 (Appendix), we have plotted the training curves for 44 different types of graphs (type: {chain,collider,jungle,full}; sizes: 3-13). For each of these, we ran 5 experiments with different seeds, which also means different ground-truth parameters, and thus different synthetic datasets. The error bars and curves are the result of averaging and computing the standard deviation across all seeds, for each of the 44 curves. In total, to generate Figure 10, we used no less than 4x11x5=220 different synthetic datasets, and succeeded each time.
> > > >
> > > > Second, the method does display an ability to “learn” an equivalence class when under-fed with interventions/data. As we mention in our second reply to Reviewer 4, and the new Figure 9 (Bottom), the algorithm quickly solves “easy” edges and non-edges, but requires much more time and data to identify the correct member within a Markov equivalence class. The figure displays the learning of a chain graph of length 10. In the example, an initial uncertainty about all edges quickly turns into uncertainty only about the direction of the edges in the chain (because, e.g., the chain with edges going in the forwards and backwards directions belong to the same Markov equivalence class). After extended training, the method converges on the correct (forward, causal) direction for those edges.
> > > >
> > > > In the visualization, this is represented by a first phase, in which the super- and sub-diagonals remain yellow while all other elements turn blue (indicating a difficulty in deciding between Markov-equivalent alternative edge directions in the super- and sub-diagonal, but also a learned confidence in the absence of edges elsewhere), followed by a second phase where the subdiagonal red-shifts (confirming the causal edges present there), while the superdiagonal alternative blue-shifts (confirming the absence of anti-causal edges)
> > > >
> > > > Had the training been ended early because of limited interventions/data after the first phase, the visualization would have appeared as in the middle of the progression of Figure 9 (Bottom), clearly indicating that the graph skeleton is linear in nature, with Markov-equivalent alternatives clearly visible but undecided upon.

---

> > > > ### Author Response · Authors · 2020-11-23
> > > > **On the Heuristic Nature and Related Work (Part 2 /3)**
> > > >
> > > > > in pg. 13, the authors say "In our setting, all variables are observed (there are no latent confounders) and all interventions are random and independent. Hence, within our setting, if the interventions are known, then the true causal graph is always identifiable in principle" This is not true. Even with known interventions, we can only talk about an equivalence class. See for example Kocaoglu et al. "Characterization and Learning of Causal Graphs with Latent Variables from Soft Interventions".
> > > >
> > > > We thank the reviewer for the reference, and have added both this citation and Jaber et al to our paper’s related-work section. There are some differences in the setup of the experiments in our paper and the one in Kocaoglu et al. Kocaoglu et al. make an important assumption of a _controlled experiment setting_, where if a variable is (soft-)intervened upon, then it is always intervened upon in the same way. Our method uses a large variety of different soft-interventions over each node, so it is not obvious that the limitation above applies to our method. Jaber et al’s paper is more applicable, since it admits a number of different intervention mechanisms for the same variable.
> > > >
> > > > > “I am still a little concerned about how the proposed method can get ALL the edges correct in multiple datasets. Even with interventional data, there is an equivalence class that you cannot go beyond without more interventions. I would be curious to see what these are for each dataset and understand if the algorithm is remaining within the equivalence class, or indeed learning something more”
> > > >
> > > > To follow up on this important question, it is important to note that our method does not currently have theoretical guarantees. However, we show that in practice, on both synthetic (chain/collider/jungle/full) and real-world datasets (BnLearn repository graphs) of various sizes, the method converges on the correct solution with few or no errors.
> > > >
> > > > In addition, we have run an additional analysis in Appendix A.13 to better understand how well our method converges. Crucially, we wanted to verify if the “heuristic” we use for learning the structural parameters can lead to the right answer if it is close enough to the true answer. Hence, we initialize the learned structural parameters $\gamma$ in the neighborhood of, but not exactly at, the optimum (ground-truth) value and we observe if the model indeed converges to the correct structure. In Figure 16, it is shown that indeed, in the neighbourhood of the optimal solution, the method converges quickly to that solution. This is evidence for the “heuristic” we use to optimize the structural parameters being aligned with the true objective (finding the ground-truth causal structure).

---

> > > > ### Author Response · Authors · 2020-11-23
> > > > **Regarding the heuristic nature of our method and identifiability proofs (Part 3/3)**
> > > >
> > > > We thank the reviewer for pointing this out, we would like to respond to the concerns regarding the heuristic nature of our method and the identifiability proofs.
> > > >
> > > > We understand that much of the past literature on causal discovery has hinged on providing identifiability proofs, and although we show that an objective function is optimized, there is no strong guarantee that the correct graph will be recovered (for example there could be a local minimum in the optimization). If the scientific community had waited for deep learning to prove that it could discover the true conditional distribution of outputs given inputs, we would not have had the progress we achieved in the last two decades in AI. We believe that it is important to take into consideration all sources of evidence about the usefulness of a method, and experimental evidence is at the heart of the success of the scientific method and should not be discarded because of an established cultural habit of relying on proofs of identifiability.
> > > >
> > > > We believe that we have addressed most of your concerns and have updated our paper with additional experiments and analysis. Would you be willing to reconsider the score of our paper based on the clarifications and updates? We greatly appreciate your time and feedback regardless.

---

> > > > ### Author Response · Authors · 2020-11-24
> > > > **Summary of Edits**
> > > >
> > > > We thank the reviewer for their feedback. Here’s a brief summary of our edits:
> > > >
> > > > - *Heuristic Nature of Gradient Estimator*: We have run more experiments to characterize the gradient estimator and verify it is attracted to the global minimum (Fig. 16).
> > > > - *Markov Equivalence Class*:  We have added visualizations that show how the method quickly discerns the Markov Equivalence Class of the problem, then slowly identifies the true causal graph within the MEC given more data (Fig. 9 Bottom).
> > > > - *Assumptions about Interventions*: We do not assume, as some papers have, the “controlled experiment setting” where a variable is always intervened upon the same way. We use data from a variety of independent interventions on each node. When sufficiently-many informative interventions are performed on well-chosen variables, the equivalence class can be made to shrink to something smaller than the Markov Equivalence Class (Pfister et al, 2019, https://arxiv.org/pdf/1911.01850.pdf), potentially enough to identify exactly the parent set of a variable.
> > > > - *Comparison to JCI* : We did more experiments to compare to JCI. We use the same experiment setup as Jaber et al (Fig. 5 & 6 in that paper’s appendix). Our proposed approach is able to recover the exact underlying causal graph (using a greater number of samples) as JCI.
> > > > - *Intervention Target Prediction Heuristic*: The target predictor we used is not an essential portion of the algorithm, and JCI could be used instead, as suggested.
> > > >
> > > > We thank you again for your valuable feedback and time reviewing this paper, and hope we have addressed most of your concerns. Would you be willing to consider a re-evaluation of our paper?

---

### Decision · Program_Chairs · 2021-01-07
**Final Decision**

**Decision:**

Reject

**Comment:**

In this paper, the authors study how to incorporate experimental data with interventions into existing pipelines for DAG learning. Mixing observational and experimental data is a well-studied problem, and it is well-known how to incorporate interventions into e.g. the likelihood function, along with theoretical guarantees and identifiability. Ultimately there was a general consensus amongst the reviewers that without additional theoretical results to advance the state of the art, the contribution of this work is limited.